# αv integrins on mesenchymal cells regulate skeletal and cardiac muscle fibrosis

I.R. Murray[1,2], Z.N. Gonzalez[2], J. Baily[2], R. Dobie[3], R.J. Wallace[1], A.C. Mackinnon[3], J.R. Smith[3], S.N. Greenhalgh[3], A.I. Thompson[3], K.P. Conroy[3], D.W. Griggs[4], P.G. Ruminski[4], G.A. Gray[5], M. Singh[4], M.A. Campbell[4], T.J. Kendall [3], J. Dai[6,7], Y. Li[6,7], J.P. Iredale[8], H. Simpson[1], J. Huard[9,10], B. Péault[2,11] & N.C. Henderson[3]

Mesenchymal cells expressing platelet-derived growth factor receptor beta (PDGFRβ) are known to be important in fibrosis of organs such as the liver and kidney. Here we show that PDGFRβ[+] cells contribute to skeletal muscle and cardiac fibrosis via a mechanism that depends on αv integrins. Mice in which αv integrin is depleted in PDGFRβ[+] cells are protected from cardiotoxin and laceration-induced skeletal muscle fibrosis and angiotensin II-induced cardiac fibrosis. In addition, a small-molecule inhibitor of αv integrins attenuates fibrosis, even when pre-established, in both skeletal and cardiac muscle, and improves skeletal muscle function. αv integrin blockade also reduces TGFβ activation in primary human skeletal muscle and cardiac PDGFRβ[+] cells, suggesting that αv integrin inhibitors may be effective for the treatment and prevention of a broad range of muscle fibroses.

[1] Department of Trauma and Orthopaedics, University of Edinburgh, Chancellors Building, Little France Campus, Edinburgh EH16 4TJ, UK. [2] BHF Centre for Vascular Regeneration & MRC Centre for Regenerative Medicine, University of Edinburgh, 5 Little France Drive, Edinburgh EH16 4UU, UK. [3] MRC Centre for Inflammation Research, The Queen's Medical Research Institute, University of Edinburgh, 47 Little France Crescent, Edinburgh EH16 4TJ, UK. [4] Center for World Health and Medicine, Saint Louis University, Edward A. Doisy Research Center, St. Louis MO 63104, USA. [5] BHF Centre for Cardiovascular Science, The Queen's Medical Research Institute, University of Edinburgh, 47 Little France Crescent, Edinburgh EH16 4TJ, UK. [6] Department of Pediatric Surgery, University of Texas McGovern Medical School, TX 77030, USA. [7] Center for Stem Cell and Regenerative Medicine, The Brown Foundation Institute of Molecular Medicine (IMM), The University of Texas Health Science Center at Houston (UT Health), TX 77030, USA. [8] University of Bristol, Senate House, Tyndall Avenue, Bristol BS8 1TH, UK. [9] Steadman Philippon Research Institute, Vail CO 81657, USA. [10] Department of Orthopaedic Surgery, University of Texas, Medical School at Houston, Houston TX 77030, USA. [11] Orthopaedic Hospital Research Center and Broad Stem Cell Research Center, University of California, Los Angeles CA 90024, USA. I.R. Murray and Z.N. Gonzalez contributed equally to this work. Correspondence and requests for materials should be addressed to B.Péa. (email: bpeault@mednet.ucla.edu) or to N.C.H. (email: Neil.Henderson@ed.ac.uk)

Skeletal and cardiac muscle fibrosis are both characterised by the excessive production and deposition of collagenous extracellular matrix by myofibroblasts, compromising myofibre contractility, tissue architecture and ultimately organ function[1–3]. Fibrosis secondary to skeletal muscle injury results in significant functional impairment and predisposes to further injury[4, 5]. Cardiac fibrosis is associated with considerable morbidity and mortality, and is a leading cause of death in industrialised countries[6]. However, the cellular and molecular mechanisms regulating fibrosis in these tissues remain poorly understood and treatment options are severely limited[6].

Iterative injury in any organ triggers a complex cascade of cellular and molecular events, including activation of extracellular matrix-producing myofibroblasts[1–3]. While this appropriate wound-healing response may be beneficial in the short term, persistence of myofibroblasts results in scar tissue formation that ultimately impairs tissue function. Within skeletal muscle, fibro-adipogenic progenitors expressing platelet-derived growth factor receptor alpha (PDGFRα) have been identified as key contributors to the myofibroblast pool in response to injury[7, 8]. Although the mesenchymal marker platelet-derived growth factor receptor beta (PDGFRβ) is increasingly recognised as labelling pro-fibrotic cells within multiple organs including liver, lung and kidney[9–11], much less is known about the PDGFRβ+ cellular compartment within skeletal and cardiac muscle. In keeping with the emerging view that fibrosis in different organs and disease states may share common cellular and molecular mechanisms, we hypothesised that PDGFRβ+ cells are also key regulators of the fibrogenic process in skeletal and cardiac muscle.

Transforming growth factor beta (TGFβ) is a key pro-fibrogenic cytokine in multiple organs including skeletal muscle and heart[12–14]. Its critical role in multiple biological processes, not least immunity and carcinogenesis, precludes pan-TGFβ blockade as a feasible therapy[15]. Therefore, the molecular pathways regulating local activation of TGFβ at the site of injury and fibrogenesis represent attractive targets for novel anti-fibrotic therapies. αv integrins have been demonstrated to play a key role in the activation of latent TGFβ1 and TGFβ3[16]. Specifically, all five αv integrins interact with a linear arginine-glycine-aspartic acid (RGD) motif present in the latency-associated peptide, which maintains TGFβ in an inactive state in the extracellular matrix. Active TGFβ can be released from the latency-associated peptide following αv integrin binding[17–20]. Furthermore, αv integrins, including integrins αvβ1, αvβ6 and αvβ8, have been shown to be key regulators of fibrogenesis in vivo in pre-clinical models of lung, liver and kidney fibrosis[9, 17, 21, 22]. However, the role of αv integrins in the regulation of muscle fibrosis has not previously been explored.

We exploited a recently developed genetic system (Pdgfrb-Cre) in mice to identify molecular mechanisms driving skeletal and cardiac muscle fibrosis, and focussed on the integrin αv subunit because of the aforementioned role of αv integrins in activating latent TGFβ, a central mediator of fibrosis. In addition to a genetic approach, we investigated the utility of a small molecule inhibitor of αv integrins (CWHM 12) as a potential anti-fibrotic therapy for both skeletal and cardiac muscle fibrosis. We established that selective depletion of αv integrins on PDGFRβ+ cells protects mice from fibrosis in both these muscle types, and αv integrin depletion on PDGFRβ+ cells had no adverse effects on skeletal muscle regeneration. Furthermore, treatment with CWHM 12 attenuates fibrosis in both skeletal muscle and heart, even after the fibrotic process is established.

## Results

### Pdgfrb-Cre targets recombination in skeletal muscle PDGFRβ+ cells.
In uninjured skeletal muscle, we found that Pdgfrb-Cre induced highly efficient recombination in PDGFRβ+ cells. This was demonstrated using double fluorescent mTmG reporter mice, which express membrane-targeted tandem dimer Tomato (tdTomato) before Cre-mediated excision and membrane-targeted green fluorescent protein (GFP) after excision[23] (Fig. 1a). Colocalisation studies of PDGFRβ-stained muscle sections from mTmG;Pdgfrb-Cre mice demonstrated faithful reporting, with 97.5% (SEM 0.46) of cells staining for PDGFRβ antibody reporting GFP, and 95.3% (SEM 1.08) of GFP+ cells staining positively for PDGFRβ (Fig. 1b, c). Virtually all GFP+ cells sorted from mTmG;Pdgfrb-Cre mouse muscle stained positively with anti-PDGFRβ antibody on flow cytometric analysis (Fig. 1d). To further assess specificity of Pdgfrb-Cre-mediated recombination, we stained uninjured skeletal muscle from mTmG;Pdgfrb-Cre mice with antibodies to CD31, ICAM-2 and CD144 (endothelial cells), CD45 (haematopoietic cells) and fast myosin (myofibres) (Supplementary Fig. 1). As expected, PDGFRβ was consistently expressed in close proximity to CD31+ endothelial cells and was not expressed by haematopoietic cells or myofibres, further confirming the specificity of Pdgfrb-Cre to mark PDGFRβ+ cells in skeletal muscle.

To evaluate the contribution of PDGFRβ+ cells to the myofibroblast pool following injury, we observed the distribution of GFP+ cells in skeletal muscle of mTmG;Pdgfrb-Cre mice following injury induced by intramuscular injection of cardiotoxin (CTX) (Fig. 1e). The field coverage of GFP+ cells and the number of GFP+ nuclei in skeletal muscle was significantly higher in mice receiving CTX injections compared to control mice receiving PBS injections, with significant differences maintained to at least 60 days post injection (Fig. 1f, g). To further characterise PDGFRβ+ cells in the skeletal muscle of mTmG; Pdgfrb-Cre mice, we isolated PDGFRβ+ reporter cells from control and injured mice. Quantitative PCR (qPCR) of mRNA obtained from viable GFP+ cells showed dramatic induction of multiple genes associated with the transition of quiescent PDGFRβ+ cells to the activated myofibroblast phenotype following CTX-induced muscle injury (Fig. 1h). These included the genes encoding PDGFRβ, alpha-smooth muscle actin (αSMA), TGFβ1, tissue inhibitor of metalloproteinase 1 (TIMP1), matrix metallopeptidase 2 (MMP2), matrix metallopeptidase 13 (MMP13), collagen 1 (Col1a1) and collagen 3 (Col3a1). Furthermore, to assess whether PDGFRβ+ cells isolated from skeletal muscle behave in a similar way to mesenchymal cells isolated from other organs and undergo transdifferentiation to a myofibroblast phenotype when plated on tissue culture plastic, we plated freshly isolated GFP+ cells from uninjured mTmG;Pdgfrb-Cre skeletal muscle on tissue culture plastic and assessed pro-fibrotic gene expression. After five days, these cells strongly expressed the myofibroblast marker αSMA (Fig. 1i) and after seven and fourteen days in culture demonstrated upregulation of myofibroblast associated genes encoding PDGFRβ, αSMA, Col1a1 and TIMP1 (Fig. 1j). Taken together, these results demonstrate that Pdgfrb-Cre efficiently targets both quiescent PDGFRβ+ cells and activated myofibroblasts in injured and fibrotic skeletal muscle.

Cells expressing CD34 and PDGFRα have been shown to label pro-fibrotic populations in skeletal muscle[7, 24, 25]. To establish where PDGFRβ+ cells stand in the hierarchy of previously recognised cell populations with pro-fibrotic potential during skeletal muscle fibrosis, we performed immunohistochemistry and flow cytometry colocalisation studies with anti-CD34 and anti-PDGFRα antibodies in uninjured and fibrotic skeletal muscle (Supplementary Fig. 2). We demonstrated 45.8% (SD 3.8) colocalisation of CD34 with PDGFRβ in uninjured skeletal muscle and 79.2% (SD 1.2) colocalisation in fibrotic skeletal muscle. We demonstrated 31.1% (SD 3.6) colocalisation of

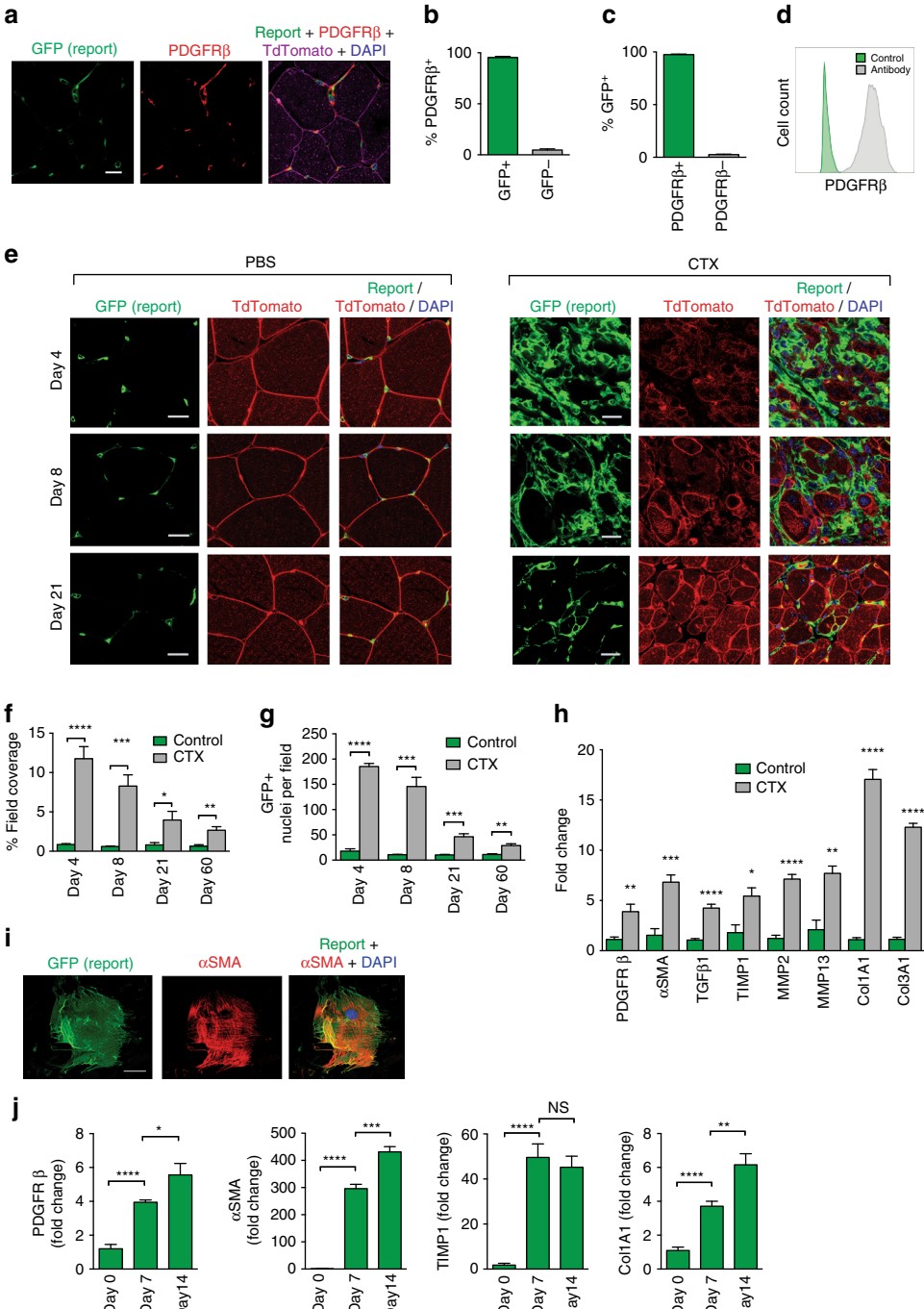

**Fig. 1** PDGFRβ-Cre effectively targets recombination in quiescent PDGFRβ[+] cells and activated myofibroblasts. **a** Immunofluorescence micrographs of skeletal muscle from mTmG;PDGFRβ-Cre mice co-stained with anti-PDGFRβ antibody. Scale bar 10 μm. **b**, **c** Quantification of GFP reporting and PDGFRβ antibody colocalisation in skeletal muscle ($n = 7$). **d** Flow cytometric analysis of PDGFRβ expression in GFP[+] cells sorted from mTmG:PDGFRβ-Cre mouse skeletal muscle ($n = 3$). **e** Immunofluorescence micrographs of skeletal muscle sections harvested from mTmG;PDGFR β-Cre reporter mice 4, 8 and 21 days after control (PBS) or CTX intramuscular injection. Scale bars 30 μm. **f** Percentage field coverage of GFP[+] cells at 4, 8, 21 and 60 days after control (PBS) or CTX intramuscular injection ($n = 4$). **g** Quantitation of GFP[+] nuclei at 4, 8, 21 and 60 days after control (PBS) or CTX intramuscular injection ($n = 4$). **h** Gene expression profile of freshly sorted GFP[+] cells from skeletal muscle at day 10 following control (PBS) or CTX intramuscular injection ($n = 4$). **i** Immunofluorescence staining of a typical GFP[+] cell sorted from uninjured skeletal muscle of mTmG;PDGFRβ-Cre reporter mice and plated on tissue culture plastic for 5 days. Scale bar 50 μm. **j** qPCR analysis of the genes encoding PDGFRβ, αSMA, Col1A1 and TIMP1 in freshly sorted GFP[+] cells from mTmG;PDGFRβ-Cre reporter mice and from GFP[+] cells cultured for 7 and 14 days ($n = 4$). Data are mean ± SEM. *$P < 0.05$, **$P < 0.01$, ***$P < 0.001$, ****$P < 0.0001$ (Student's $t$-test).

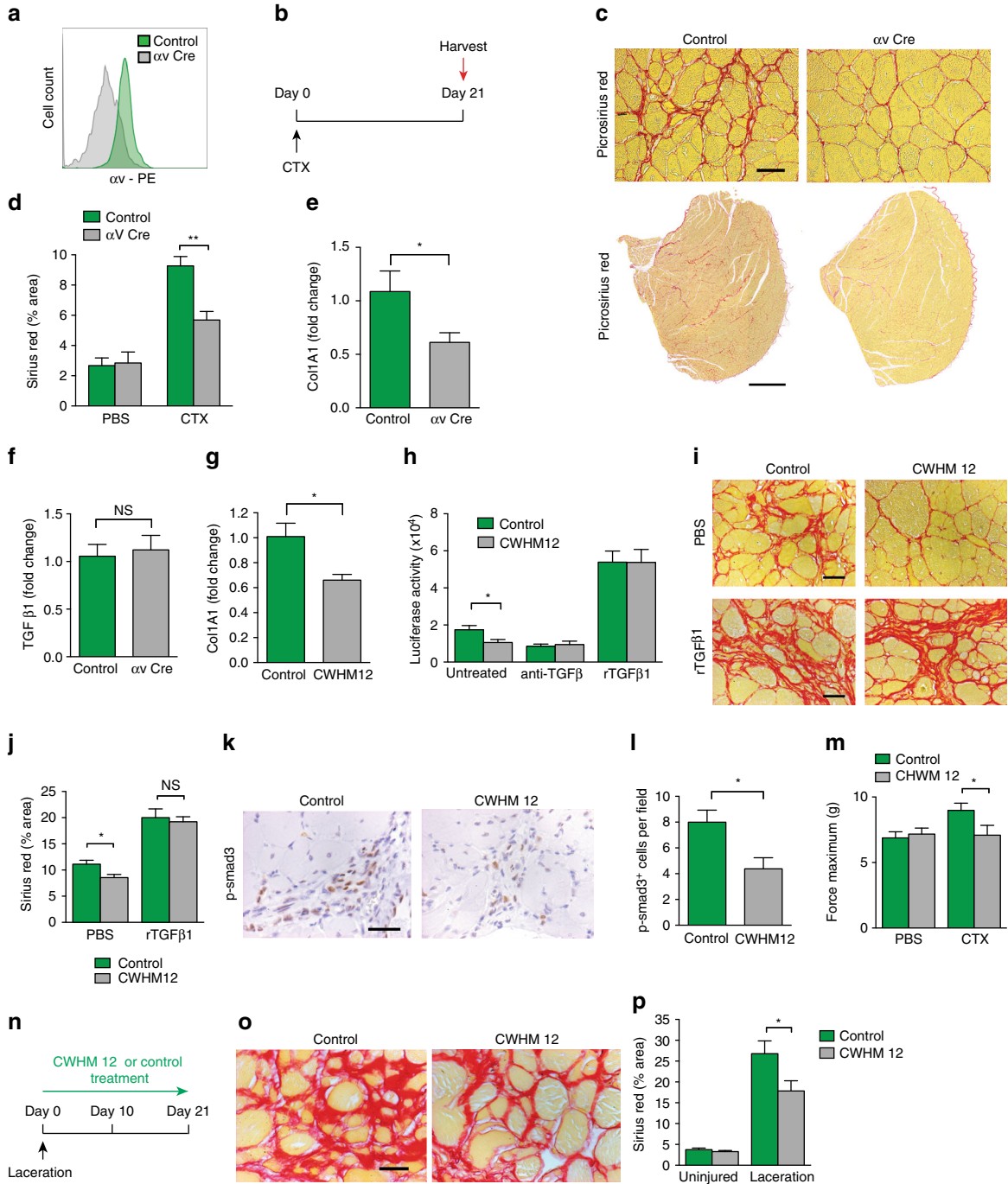

**Fig. 2** αv integrins on PDGFRβ[+] cells regulate skeletal muscle fibrosis via TGFβ activation. **a** αv expression on PDGFRβ[+] cells from control and *Itgav*[flox/flox]; PDGFRβ-Cre (αv Cre) mice using flow cytometry (*n* = 3). **b** αv Cre and control mice received intramuscular CTX with muscles harvested at day 21. **c** Representative images of picrosirius red stained sections 21 days after CTX. Upper panels: scale bar 25 μm, Lower panels: 400 μm. **d** Digital image analysis quantification of picrosirius red staining in control and αv Cre mice day 21 post CTX or control injection (*n* = 7). **e** qPCR of *Col1a1* in control and αv Cre PDGFRβ[+] cells culture-activated for 5 days (*n* = 7). **f** qPCR of TGFβ1 in control and αv Cre PDGFRβ[+] cells culture-activated for 5 days (*n* = 7). **g** qPCR of *Col1a1* in GFP[+] cells from mTmG;PDGFRβ-Cre mice cultured with CWHM 12 and control for 5 days (*n* = 3). **h** TGFβ activation by control- or CWHM 12-treated GFP[+] cells from mTmG;PDGFRβ-Cre mice (*n* = 6). TGFβ activation was assessed alone, with TGFβ–blocking antibody (Anti–TGFβ), and with recombinant human TGFβ1 (rhTGFβ). **i** Representative images of picrosirius red stained sections from mice treated with CWHM12 or control from the time of CTX. Mice received rhTGFβ1 (lower panels) to the region of injured muscle every 72 h up to 21 days. **j** Digital image analysis quantitation of picrosirius red stained sections from CWHM12 or control treated mice that received rhTGFβ1 or PBS to the region of injured muscle (*n* = 5). **k** Representative images of phospho-SMAD 3 stained sections from control or CWHM12-treated mice. Scale bar 30 μm. **l** Quantitation of phospho-SMAD 3 stained sections from mice treated with control or CWHM12 from the time of CTX (*n* = 10). **m** Force required for passive elongation of muscles 21 days following intramuscular CTX in mice treated with CWHM 12 or control (*n* = 9). **n** Laceration-induced skeletal muscle fibrosis model dosing regime. **o** Representative images of picrosirius red stained sections from control and CWHM 12 treated mice. Scale bar 25 μm. **p** Digital image analysis quantitation of picrosirius red staining in CWHM12 and control treated mice (*n* = 10). Data are expressed as mean ± SEM. *$P < 0.05$, **$P < 0.01$ (Student's *t*-test)

PDGFRα with PDGFRβ in uninjured skeletal muscle and 44.8% (SD 1.2) colocalisation in injured skeletal muscle. These data demonstrate that the pro-fibrotic PDGFRβ[+] cell population significantly overlaps with the previously described pro-fibrotic populations in skeletal muscle.

**αv integrins on PDGFRβ[+] cells regulate skeletal muscle fibrosis.** To establish the utility of *Pdgfrb*-Cre to identify molecular mechanisms driving skeletal muscle fibrosis, we focussed on the integrin αv subunit because of the role of αv-containing integrins in activating latent TGFβ, a central mediator of fibrosis[17–21, 26, 27]. We used flow cytometry to assess αv integrin expression levels on different cell lineages within uninjured skeletal muscle. In addition to the high expression levels observed on PDGFRβ[+] cells, we also found expression of αv integrins on CD31[+], CD34[+] and PDGFRα[+] populations in skeletal muscle (Supplementary Fig. 3a–d). Flow cytometry, using an antibody raised against a portion of the αv extracellular domain, revealed that αv integrin was efficiently depleted from *Itgav*[flox/flox];*Pdgfrb*-Cre PDGFRβ[+] cells in skeletal muscle (Fig. 2a). To induce skeletal muscle fibrosis, *Itgav*[flox/flox];*Pdgfrb*-Cre and control mice received intramuscular CTX, with muscles harvested at day 21 for analysis as previously described[28] (Fig. 2b). *Itgav*[flox/flox];*Pdgfrb*-Cre mice were significantly protected from CTX-induced skeletal muscle fibrosis, as determined by picrosirius red staining for collagen and digital morphometry (Fig. 2c, d). In keeping with previous studies[29–31], differentiating and regenerating myofibres in skeletal muscle expressed αSMA, and therefore this marker was not used in the quantification of fibrosis in in vivo sections. To assess whether *itgav*[flox/flox];PDGFRβ-cre mice had an aberrant immune response to CTX we assessed neutrophil and macrophage infiltration in skeletal muscle from CTX treated *itgav*[flox/flox];PDGFRβ-cre and control mice. There was no difference in the degree of initial injury or inflammatory infiltrates (Supplementary Fig. 4a–c), or differences in muscle regeneration following CTX, as assessed by Evans blue extravasation at day 8 post CTX injection[7] (Supplementary Fig. 4d). Furthermore, depletion of αv integrins on PDGFRβ[+] cells did not affect neovascularisation following skeletal muscle injury (Supplementary Fig. 4e) between the two genotypes. These data demonstrate that *Pdgfrb*-Cre effectively targets gene deletion to skeletal muscle PDGFRβ[+] cells during fibrogenesis, and that αv integrins in PDGFRβ[+] cells regulate CTX-induced skeletal muscle fibrosis.

Collagen 1 is the major fibrillar collagen deposited in the extracellular matrix during skeletal muscle fibrosis[8]. To determine whether loss of αv integrins affected induction of *Col1a1* gene expression, control and αv-null (αv Cre) skeletal muscle PDGFRβ[+] cells were activated in culture for five days. *Col1a1* expression was significantly reduced in α-null PDGFRβ[+] cells compared to control (Fig. 2e). TGFβ1 is a major pro-fibrogenic cytokine and a potent inducer of collagen gene expression and myofibroblast transdifferentiation[12]. Therefore, we assessed TGFβ1 mRNA levels in control and αv Cre PDGFRβ[+] cells, and found similar levels between the two groups, demonstrating that the reduction in *Col1a1* expression in αv Cre PDGFRβ[+] cells is not secondary to a decrease in TGFβ1 mRNA expression (Fig. 2f).

We then assessed the effect of a small molecule inhibitor of αv integrins, CWHM 12, and its control enantiomer (CWHM 96) on *Col1a1* gene expression in skeletal muscle PDGFRβ[+] cells activated in culture. CWHM 12 is a synthetic small-molecule RGD peptidomimetic antagonist that consists of a cyclic guanidine-substituted phenyl group as the arginine mimetic and a phenyl-substituted beta amino acid as the aspartic acid mimetic, both linked by glycine[9]. CWHM 96 is the R enantiomer of CWHM 12 and differs only in the orientation of its carboxyl

(CO2H) groups. In previous studies, CWHM 12, but not the control enantiomer CWHM 96, demonstrated high potency against αv integrins in in vitro ligand-binding assays[9]. Treatment with CWHM 12, but not control (CWHM 96), inhibited *Col1a1* expression in skeletal muscle PDGFRβ[+] cells in culture (Fig. 2g). Furthermore, co-culture of control and CWHM 12-treated PDGFRβ[+] cells with mink lung epithelial reporter cells (TMLCs), expressing firefly luciferase under the control of the TGFβ-sensitive plasminogen activator inhibitor promoter[32], demonstrated a significant decrease in TGFβ activation by CWHM 12-treated cells compared to control (Fig. 2h). This difference was eliminated by TGFβ blocking antibody or addition of activated TGFβ1 (Fig. 2h).

Our data presented above suggested that inhibition of αv integrins on PDGFRβ[+] cells represents a valuable therapeutic strategy in skeletal and cardiac muscle fibrosis. Therefore, we used the small molecule αv integrin inhibitor CWHM 12 to determine whether pharmacologic blockade of αv integrins could attenuate skeletal muscle fibrosis. Mice were treated with the small molecule inhibitor of αv integrins (CWHM 12) or control from the time of CTX injury. Treatment with CWHM 12 significantly reduced skeletal muscle fibrosis, as determined by picrosirius red staining for collagen (Fig. 2i). To further investigate whether the anti-fibrotic effect of αv integrin blockade is secondary to reduced TGFβ activation in vivo, we administered recombinant TGFβ1 every 72 h to mice receiving either CWHM 12 or control post CTX injection. Administration of active recombinant TGFβ1 rescued the anti-fibrotic effect of CWHM 12 (Fig. 2i, j). Furthermore, as we had shown that CWHM 12-treated PDGFRβ[+] cells are defective in their ability to activate TGF-β in vitro, we extended our findings in vivo by examining canonical TGF-β signalling by immunostaining for phosphorylated SMAD3 (p-Smad3) (Fig. 2k, l). Digital image quantification demonstrated significantly reduced p-smad3 signalling in skeletal muscles of mice treated with CWHM 12 compared to control following intramuscular CTX injection (Fig. 2k, l). Taken together, these results strongly suggest that the protection from skeletal muscle fibrosis observed following αv integrin blockade is at least in part a consequence of reduced TGF-β activation.

In the setting of skeletal muscle fibrosis, stiffness 'contractures' result in significant morbidity and are a major cause of loss of muscle function. We therefore used commercially available, customized muscle physiology testing equipment to assess muscle stiffness following CTX-induced injury, to provide functional data with direct clinical relevance. We found that muscle stiffness was significantly reduced in mice treated with CWHM 12 compared to control (Fig. 2m), demonstrating that αv integrin blockade using CWHM 12 can improve muscle function following CTX-induced muscle injury.

In order to investigate whether αv integrin-mediated regulation of skeletal muscle fibrosis is common to different modes of skeletal muscle injury, we used a laceration model of skeletal muscle fibrosis[33, 34]. Pharmacologic blockade of αv integrins using CWHM 12 delivered via osmotic minipumps protected mice from laceration-induced fibrosis (Fig. 2n–p), demonstrating that αv integrin blockade is anti-fibrotic in skeletal muscle models with very different mechanisms of injury.

To further investigate the mechanisms through which αv integrin inhibition reduces skeletal muscle fibrosis, we investigated the effects of αv integrin blockade on the migration and apoptosis of PDGFRβ[+] cells isolated from skeletal muscle. We found that pharmacologic blockade of αv integrins had no effect on migration or apoptosis of PDGFRβ[+] cells isolated from skeletal muscle (Supplementary Fig. 4f–h).

The αv integrin has five possible β subunit binding partners (β1, β3, β5, β6 and β8), each of which has been reported to bind

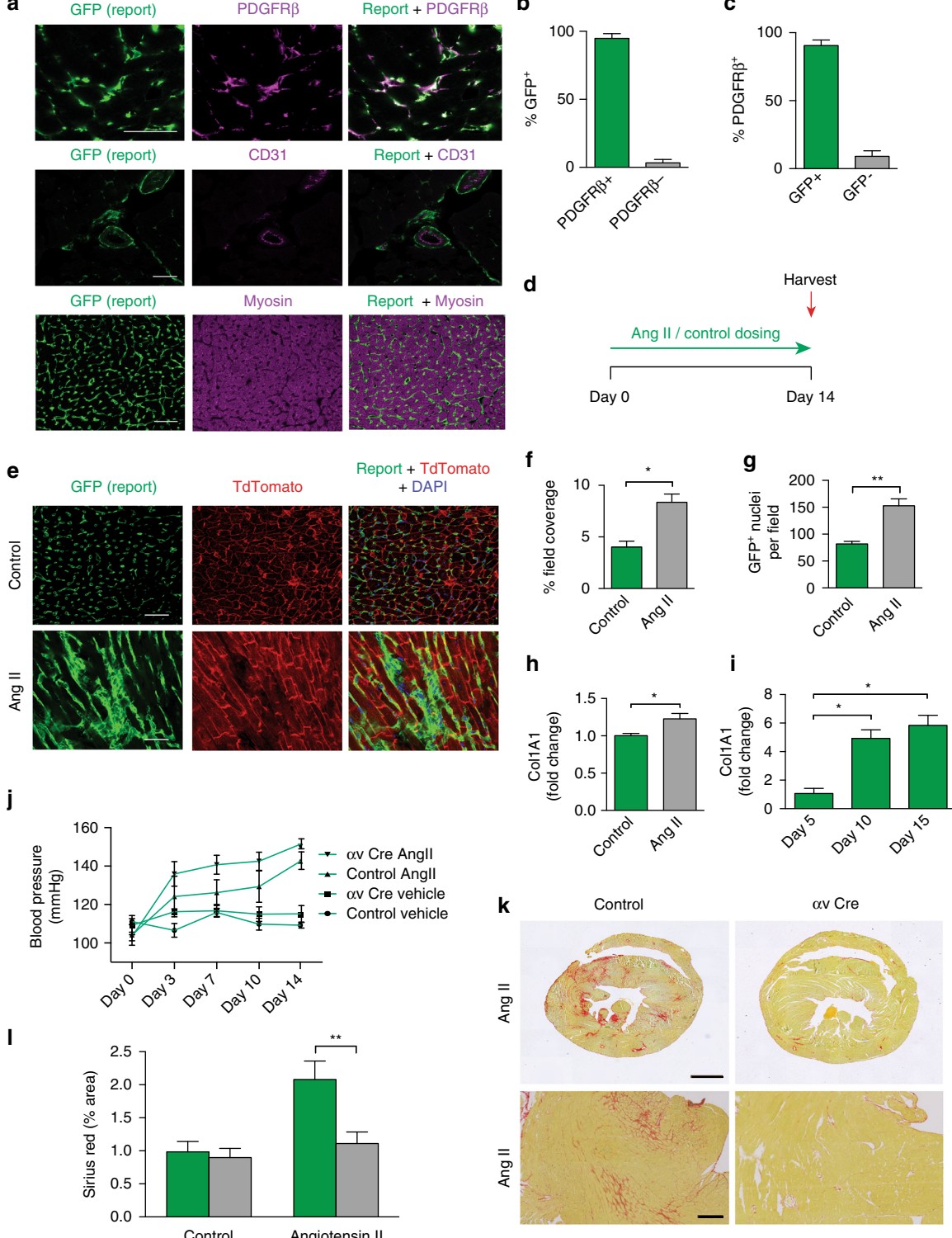

**Fig. 3** Selective depletion of αv integrins on PDGFRβ[+] cells is protective in cardiac fibrosis. **a** Immunofluorescence micrographs of heart muscle from mTmG;PDGFRβ-Cre mice co-stained with anti-PDGFRβ (scale bar 5 μm), anti-CD31 (scale bar 5 μm) and anti-myosin antibodies (scale bar 30 μm). **b**, **c** Quantification of GFP reporting and PDGFRβ antibody staining colocalisation in cardiac muscle (n = 6). **d** Overview of the cardiac fibrosis model. Mice were treated with 200 ng kg$^{-1}$ min$^{-1}$ AngII or vehicle control for 14 days prior to harvest and analysis of tissues. **e** Immunofluorescence micrographs of heart sections from control or AngII-treated mTmG;PDGFRβ-Cre mice. Scale bars 30 μm. **f** Percentage field coverage of PDGFRβ[+] cells at day 14 in vehicle control or AngII-treated mTmG;PDGFRβ-Cre mice (n = 3). **g** Quantitation of GFP[+] nuclei at day 14 in vehicle control or AngII-treated mTmG;PDGFRβ-Cre mice (n = 4) **h** qPCR of *Col1a1* in cardiac PDGFRβ[+] cells sorted from AngII injured and control hearts (n = 5) **i** qPCR of *Col1a1* in cardiac PDGFRβ[+] cells activated in culture for 5 days (n = 3). **j** Blood pressure response of control and *Itgav*[flox/flox];PDGFRβ-Cre (αv Cre) mice to AngII treatment or vehicle (n = 6). **k** Picrosirius red staining of cardiac tissue 14 days after commencement of AngII treatment in control and *Itgav*[flox/flox];PDGFRβ-Cre mice. Scale bars 1 mm in whole heart sections, 70 μm for magnified fields. **l** Digital image analysis of collagen staining (n = 6 mice per group). Data are expressed as mean ± SEM. *P < 0.05, **P < 0.01 (Student's *t*-test)

and/or activate latent TGFβ. We investigated expression of the various αv integrin β subunits in PDGFRβ$^+$ reporter cells isolated from control and fibrotic skeletal muscle (Supplementary Fig. 3e). We found expression of the genes encoding β$_1$, β$_3$, β$_5$ and β$_8$ on PDGFRβ$^+$ reporter cells, with significant increases in β$_1$, β$_5$ and β$_8$. As expected, we did not detect expression of the epithelial-restricted integrin β$_6$.

**Selective αv integrins depletion inhibits cardiac fibrosis.** Cardiac myofibroblasts are central mediators of extracellular matrix deposition during cardiac fibrogenesis and have previously been shown to express high levels of PDGFRβ[11, 27]. We therefore hypothesised that *Pdgfrb*-Cre marks myofibroblasts in the heart and allows manipulation of specific genes in these cells. We initially examined the hearts of mTmG;*Pdgfrb*-Cre reporter mice to confirm the specificity of *Pdgfrb*-Cre-induced reporting. As in skeletal muscle, PDGFRβ staining demonstrated a high degree of colocalisation with GFP$^+$ reporter cells (Fig. 3a–c). To further assess specificity of *Pdgfrb*-Cre-mediated recombination, we stained uninjured cardiac muscle from mTmG;*Pdgfrb*-Cre mice with antibodies to CD31, ICAM-2 and CD144 (endothelial cells), CD45 (haematopoietic cells) and fast myosin (myofibres). As expected, GFP$^+$ reporter cells were consistently found in close proximity to CD31$^+$ endothelial cells (Fig. 3a, middle panel; Supplementary Fig. 5a, b). GFP was not expressed by haematopoietic cells (Supplementary Fig. 5c) or myofibres (Fig. 3a, lower panel), further confirming the specificity of *Pdgfrb*-Cre to mark PDGFRβ$^+$ cells in cardiac muscle. To examine the role of PDGFRβ$^+$ cells in cardiac fibrosis, we administered angiotensin II (AngII) to mTmG;*Pdgfrb*-Cre mice for 14 days (Fig. 3d). After 14 days of AngII treatment, GFP$^+$ reporter cells were distributed between myofibres and throughout the interstitium in a pattern characteristic for cardiac myofibroblasts (Fig. 3e). Field coverage of GFP$^+$ reporter cells, and the number of GFP$^+$, DAPI$^+$ nuclei, were significantly increased after 14 days of AngII treatment (Fig. 3f, g). Expression of the pro-fibrotic gene *Col1a1* was significantly increased in PDGFRβ$^+$ cells isolated from fibrotic hearts (Fig. 3h). Furthermore, freshly isolated GFP$^+$ cells from mTmG;*Pdgfrb*-Cre mouse hearts activated in culture demonstrated upregulation of *Col1a1* in keeping with transdifferentiation to an activated myofibroblast phenotype (Fig. 3i).

Populations of cells expressing the markers PDGFRα and CD34 have been implicated in the development of cardiac fibrosis[35, 36]. To establish where PDGFRβ$^+$ cells stand in the hierarchy of previously recognised cell populations with pro-fibrotic potential in the heart, we performed flow cytometry and immunohistochemistry colocalisation studies with anti-CD34 and anti-PDGFRα antibodies in healthy and fibrotic hearts. Using flow cytometry, we demonstrated 49.1% (SD 2.3) colocalisation of CD34 with PDGFRβ in uninjured cardiac muscle and 51.5% (SD 8.8) colocalisation in fibrotic cardiac muscle. We demonstrated 44.8% (SD 1.2) colocalisation of PDGFRα with PDGFRβ in uninjured cardiac muscle and 71.6% (SD 4.7) colocalisation in fibrotic hearts (Supplementary Fig. 6). These data demonstrate that the pro-fibrotic PDGFRβ$^+$ cell population significantly overlaps with the previously described pro-fibrotic populations in the heart.

To investigate whether *Pdgfrb*-Cre-mediated depletion of αv integrins on PDGFRβ$^+$ cells protects against cardiac fibrosis, we treated control or *Itgav*$^{flox/flox}$;*Pdgfrb*-Cre mice with vehicle control or AngII, and harvested hearts after 14 days. Serial blood pressure measurements confirmed that control and *Itgav*$^{flox/flox}$;*Pdgfrb*-Cre mice displayed an appropriate hypertensive response to AngII treatment (Fig. 3j). *Itgav*$^{flox/flox}$;*Pdgfrb*-Cre mice were significantly protected from AngII-induced cardiac fibrosis, as

determined by collagen staining and digital morphometric assessment (Fig. 3k, l). The αv integrin has five possible β subunit binding partners (β$_1$, β$_3$, β$_5$, β$_6$ and β$_8$), each of which has been reported to bind and/or activate latent TGFβ. We investigated expression of the various αv integrin β subunits in PDGFRβ$^+$ reporter cells isolated from control and fibrotic cardiac muscle (Supplementary Fig. 7a). We found expression of the genes encoding β$_1$, β$_3$, β$_5$ and β$_8$ on PDGFRβ$^+$ reporter cells, with a significant decrease in β$_1$ and a significant increase in β$_3$ in PDGFRβ$^+$ reporter cells isolated from fibrotic hearts. As expected, we did not detect expression of the epithelial-restricted integrin β$_6$. These data demonstrate that αv integrin depletion on PDGFRβ$^+$ cells also protects against AngII-induced cardiac fibrosis.

To investigate whether αv integrin depletion influences cardiac hypertrophy we analysed cardiomyocyte cross-sectional area in hearts from control and *Itgav*$^{flox/flox}$;*Pdgfrb*-Cre mice treated for 14 days with AngII. We found no difference in cross-sectional area of cardiomyocytes between control or *Itgav*$^{flox/flox}$;*Pdgfrb*-Cre mice following AngII treatment (Supplementary Fig. 8a, b).

To further investigate the mechanisms through which αv integrin inhibition reduces heart fibrosis, we investigated the effects of αv integrin blockade on the migration and apoptosis of PDGFRβ$^+$ cells isolated from cardiac muscle. We found that pharmacologic blockade of αv integrins had no effect on migration or apoptosis of PDGFRβ$^+$ cells isolated from cardiac muscle (Supplementary Fig. 9a, b). In addition, depletion of αv integrins on PDGFRβ$^+$ cells did not affect neovascularisation of the fibrotic heart (Supplementary Fig. 9c).

We used flow cytometry to assess αv integrin expression levels on different cell lineages within uninjured cardiac muscle. In addition to the high expression levels observed on PDGFRβ$^+$ cells, we also found expression of αv integrins on CD31$^+$, CD34$^+$ and PDGFRα$^+$ populations in cardiac muscle (Supplementary Fig. 7b–e).

**αv integrins blockade attenuates skeletal muscle and cardiac fibrosis.** Our data presented above suggested that inhibition of αv integrins on PDGFRβ$^+$ cells represents a valuable therapeutic strategy in skeletal and cardiac muscle fibrosis. Therefore, we next asked whether CWHM 12 can attenuate skeletal muscle and cardiac fibrosis once established. Mice were injected with CTX and then 10 days later Alzet pumps containing either CWHM 12 or control (CWHM 96) were inserted for the final 11 days of our protocol (Fig. 4a). Treatment with CWHM 12 significantly reduced skeletal muscle fibrosis compared to control even after fibrotic disease had been established (Fig. 4b, c).

We then examined whether CWHM 12 could prevent progression of established cardiac fibrosis. Mice were treated with AngII via Alzet osmotic minipumps for seven days prior to insertion of Alzet minipumps containing either CWHM 12 or control (CWHM 96). Mice then received AngII and either CWHM 12 or CWHM 96 for a further 7 days (Fig. 4d). Treatment with CWHM 12 significantly reduced cardiac fibrosis compared to control even after fibrotic disease had been established (Fig. 4e, f). Therefore, our findings suggest that inhibition of αv integrins by the small molecule CWHM 12 has clinical utility in the treatment of patients with skeletal muscle or cardiac fibrosis.

In order to determine the plasma and tissue concentrations in skeletal and cardiac muscle of CWHM 12 and the control enantiomer CWHM 96, we implanted osmotic minipumps in C57/BL6 mice and delivered the compounds at a dose of 100 mg kg$^{-1}$ day$^{-1}$. Mice were sacrificed at day 3 when the drug had reached steady state, and terminal plasma, tibialis anterior and

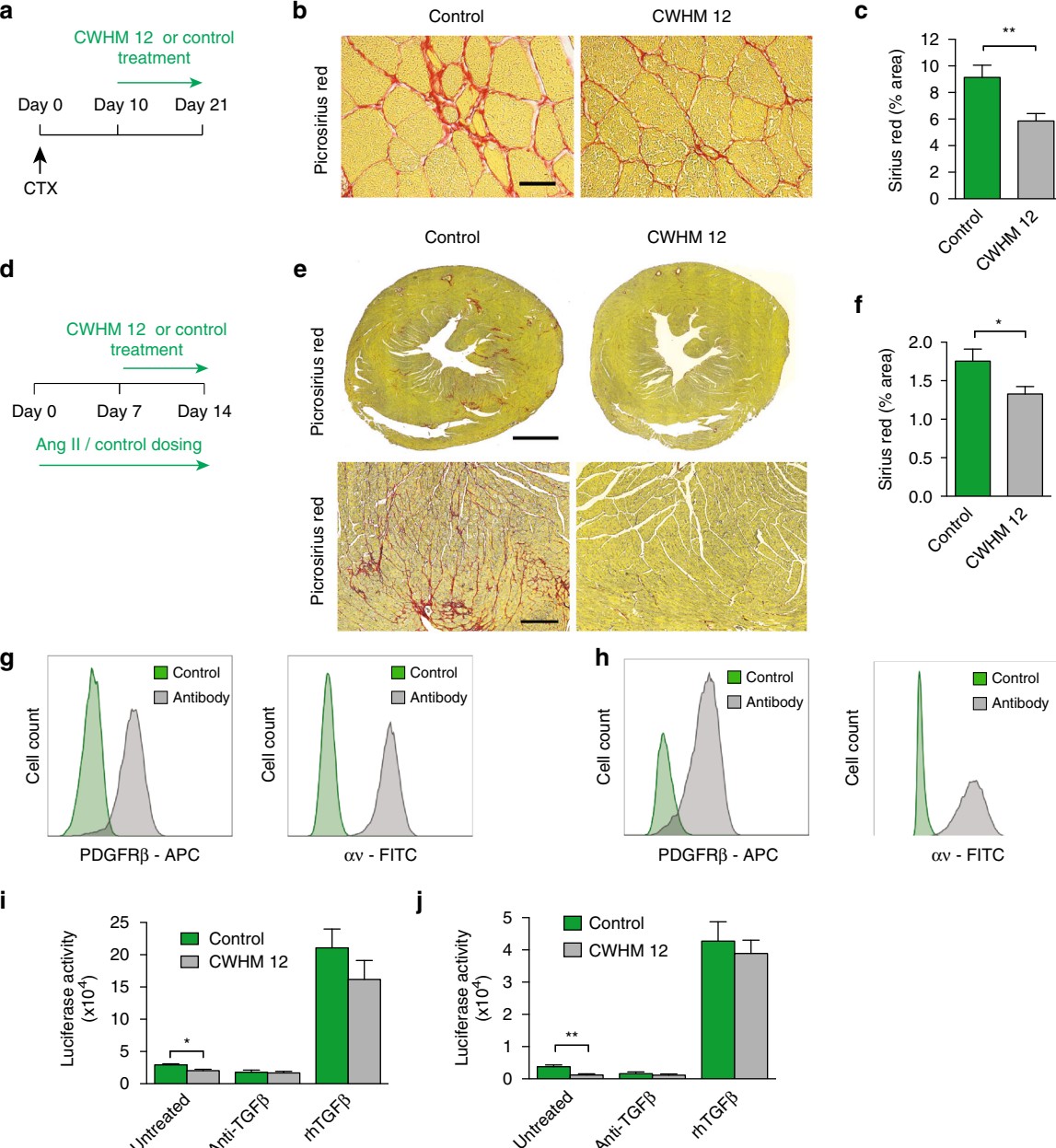

**Fig. 4** Blockade of αv integrins by a small molecule (CWHM 12) attenuates established skeletal muscle and cardiac fibrosis, and αv integrins represent a tractable therapeutic target in human muscle fibrosis. **a** Dosing regime in the therapeutic skeletal muscle fibrosis model. Alzet osmotic minipumps containing CWHM 12 or CWHM 96 (control) were inserted ten days after intramuscular CTX injection. Tissues were harvested at day 21 following CTX injection. **b** Representative images of picrosirius red stained sections from control- and CWHM 12-treated mice. Scale bar 25 μm. **c** Digital image analysis quantification of collagen (picrosirius red staining) ($n = 10$). **d** Dosing regime in the therapeutic cardiac fibrosis model. Seven days following commencement of AngII treatment, Alzet osmotic minipumps containing CWHM 12 or CWHM 96 (control) were inserted. Tissues were harvested at day 14 after commencement of AngII treatment. **e** Representative images of picrosirius red stained sections from control- and CWHM 12-treated mice. Scale bars 1 mm in whole heart sections, 70 μm for magnified fields. **f** Digital image analysis quantification of collagen (picrosirius red staining) ($n = 11$). **g**, **h** Flow cytometric analysis of PDGFRβ and αv integrin expression on PDGFRβ+ cells from human skeletal muscle (**g**) and heart (**h**). **i**, **j** TGFβ activation by control- or CWHM 12-treated PDGRβ+ cells isolated from human skeletal muscle (**i**) and cardiac muscle (**j**) ($n = 4$). TGFβ activation was assessed alone, in the presence of TGFβ-blocking antibody (clone 1D11, 40 μg ml⁻¹) (anti-TGFβ), and in the presence of recombinant human TGFβ1 (rhTGFβ) (300 pg ml⁻¹). Data are expressed as mean ± SEM. *$P < 0.05$, **$P < 0.01$ (Student's *t*-test)

cardiac muscle samples were taken for compound analysis using liquid chromatography–mass spectrometry (LC–MS/MS). The mean CWHM 12 plasma concentration was almost identical to what we have reported in previous minipump studies using these compounds[37]. Furthermore, CWHM 96 plasma levels were significantly higher than CWHM 12 levels, in keeping with

previous studies[37]. Drug enantiomers frequently exhibit differing pharmacokinetic parameters because a number of mechanisms can be stereoselective, among them target interaction, metabolic clearance, renal clearance and protein binding. For each compound, the tissue concentrations measured in skeletal and cardiac muscle were similar to each other (Supplementary

Fig. 10). These data demonstrate that the small molecule αv integrin inhibitor (CWHM 12) was present in plasma, skeletal muscle and cardiac muscle at the expected concentrations.

**αv integrins on PDGFRβ⁺ cells are a potential therapeutic target.** To further explore whether αv integrins represent a tractable therapeutic target in human skeletal muscle and cardiac fibrosis, we sorted PDGFRβ⁺ cells from human skeletal and cardiac muscle. Flow cytometric analysis of PDGFRβ⁺ cells isolated from human skeletal muscle demonstrated that the vast majority of these cells also expressed αv integrins (Fig. 4g). Co-culture of CWHM 12- or control- (CWHM 96) treated human skeletal muscle PDGFRβ⁺ cells with TMLCs demonstrated a significant reduction in TGFβ activation by PDGFRβ⁺ cells treated with CWHM 12 compared to control (Fig. 4i). This difference was eliminated by addition of TGFβ-blocking antibody or by addition of activated TGFβ1 (Fig. 4i). Furthermore, flow cytometric analysis of PDGFRβ⁺ cells isolated from human heart demonstrated that the vast majority of these cells also expressed αv integrins (Fig. 4h). Co-culture of CWHM 12- and control- (CWHM 96) treated human cardiac PDGFRβ⁺ cells with TMLCs demonstrated a significant reduction in TGFβ activation by PDGFRβ⁺ cells treated with CWHM 12 compared to control (Fig. 4j). This difference was eliminated by addition of TGFβ-blocking antibody or addition of activated TGFβ1 (Fig. 4j). Taken together, these data demonstrate that αv integrins are highly expressed and targetable on PDGFRβ⁺ cells in both human skeletal and cardiac muscle, and therefore represent a potential novel therapeutic target in the treatment of skeletal muscle and cardiac fibrosis.

## Discussion

Muscle fibroses, including skeletal and cardiac muscle fibrosis, are major causes of morbidity and mortality worldwide, however treatment options for these conditions remain severely limited[6, 38]. Therefore, in order to aid the design of rational, targeted, anti-fibrotic therapies to treat patients with skeletal muscle or cardiac fibrosis, we used *Pdgfrb*-Cre to help identify the key molecular mechanisms driving muscle fibrosis. We found that *Pdgfrb*-Cre effectively targeted quiescent PDGFRβ⁺ cells and activated myofibroblasts in both skeletal and cardiac muscle. Selective depletion of αv integrins on PDGFRβ⁺ cells protected mice from fibrosis in both these muscle types, and αv integrin depletion on PDGFRβ⁺ cells had no adverse effects on skeletal muscle regeneration, underlining the potential therapeutic utility of this approach to treat muscle fibrosis. Furthermore, treatment with a small-molecule inhibitor of αv integrins (CWHM 12) attenuated fibrosis in both skeletal muscle and heart, even after the fibrotic process had become established, suggesting that inhibition of this pathway may be effective in a clinical setting, where patients typically present with pre-existing muscle fibrosis. In addition, we demonstrated that skeletal muscle stiffness was significantly reduced in mice treated with CWHM 12 compared to control, demonstrating that αv integrin blockade using CWHM 12 can improve muscle function following CTX-induced muscle injury.

We found that treatment with CWHM 12 inhibited Col1a1 expression in skeletal muscle PDGFRβ⁺ cells in culture. Furthermore, CWHM 12 treatment, in both primary mouse and human PDGFRβ⁺ cells isolated from skeletal and cardiac muscle, demonstrated a significant decrease in TGFβ activation compared to control, suggesting that αv integrins on PDGFRβ⁺ cells regulate skeletal and cardiac muscle fibrosis, at least in part, via the activation of TGFβ. Targeted blockade of TGFβ1 signalling during muscle fibrogenesis, through inhibition of αv integrins with a

small molecule such as CWHM 12, may yield the desired anti-fibrotic effects without the unwanted, potentially deleterious side effects of pan-TGFβ blockade, such as induction of autoimmunity or carcinogenesis[15].

Myofibroblasts are the major source of extracellular matrix proteins during muscle fibrogenesis[4], and a number of putative myofibroblast progenitor populations have been implicated in the development of muscle fibrosis, including fibro-adipogenic progenitors[7, 8, 24], and cells expressing Gli1[11], ADAM12[28] and PDGFRα[8]. However, despite PDGFRβ⁺ mesenchymal cells being extensively studied in the context of solid organ fibrosis, much less is known about the role of these mesenchymal cells in the regulation of muscle fibrogenesis. We saw significant expansion of PDGFRβ⁺ cells in both cardiotoxin-induced skeletal muscle fibrosis and AngII-induced model of cardiac fibrosis, suggesting that proliferation of PDGFRβ⁺ cells is a conserved injury response in different types of muscle, regardless of the initial mode of muscle injury. In keeping with widespread use of PDGFRβ as a pericyte marker[39, 40], *Pdgfrb*-Cre used in the present study labelled perivascular cells in uninjured skeletal and cardiac muscle. However, the co-labelling of PDGFRβ with PDGFRα⁺ and CD34⁺ populations indicates that the pro-fibrotic PDGFRβ⁺ cell population in skeletal and cardiac muscle significantly overlaps with the previously described pro-fibrotic cells including fibro-adipogenic progenitors and cardiac fibroblasts[7, 8, 35, 36].

Of note a recent paper by Guimarães-Camboa et al.[41] suggested that *Pdgfrb*-Cre extensively labels multiple cellular lineages including endothelial and haematopoietic lineages in cardiac and skeletal muscle. To investigate the discrepancy between our data using *Pdgfrb*-Cre in skeletal and cardiac muscle and the data published by Guimarães-Camboa et al., we stained skeletal and cardiac muscle from mTmG;*Pdgfrb*-Cre reporter mice with the hematopoietic marker CD45 and found no colocalisation between PDGFRβ reporter cells and CD45 (Supplementary Figs. 1 and 5). We also stained skeletal and cardiac muscle from mTmG;*Pdgfrb*-Cre reporter mice with the endothelial markers CD31, CD144 and ICAM-2 and demonstrated no colocalisation of PDGFRβ reporter cells with CD31, CD144 and ICAM-2 (Supplementary Fig. 1 and 5). Furthermore, data published by Guimarães-Camboa et al.[41] suggested that recombination of myofibres was widespread in skeletal muscle in *Pdgfrb*-Cre reporter mice. However, in marked contrast we have not seen widespread recombination of myofibres in *Pdgfrb*-Cre reporter mice, as evidenced by 97.5% (SEM 0.46) of cells staining for PDGFRβ reporting GFP, and 95.3% (SEM 1.08) of GFP⁺ cells staining positively for PDGFRβ (Fig. 1b, c). Virtually all GFP⁺ cells sorted from mTmG;*Pdgfrb*-Cre mouse skeletal muscle stained positively with anti-PDGFRβ antibody on flow cytometric analysis (Fig. 1d). Furthermore, we demonstrate a lack of colocalisation of PDGFRβ reporter cells with anti-myosin staining in skeletal muscle of *Pdgfrb*-Cre reporter mice (Supplementary Fig. 1), and as evidenced by multiple images throughout our article we do not see widespread recombination of myofibres in skeletal muscle of PDGFRβ Cre reporter mice.

Functional heterogeneity within skeletal and heart muscle mesenchymal cells is increasingly being recognised. It is highly likely that in the coming years distinct mesenchymal sub-populations and subclasses will be identified and characterised that are the major regulators of muscle injury, fibrosis and regeneration. Ever-increasing fidelity in the elucidation of the key functional subpopulations of muscle mesenchymal cells will hopefully facilitate the discovery of highly targeted, potent, anti-fibrotic treatments. Ideally this will allow therapeutic inhibition of the pro-fibrogenic subpopulations, whilst not perturbing the function of the mesenchymal populations involved in normal tissue homeostasis and regeneration.

# Methods

**Mice.** mTmG (TdTomato-GFP)[23] were obtained from the Jackson Laboratory and crossed with *Pdgfrb*-Cre mice[42] provided by R. Adams. *Itgav*[flox/flox] mice[43] were obtained from A. Lacy-Hulbert, and were maintained on C57BL/6 background. Genotyping of mice was performed by PCR. Wild type C57BL/6 mice were purchased from Charles River. Mice used for experiments were males aged 8–12 weeks and were housed under specific pathogen-free conditions. All animal work was approved by the UK Home Office, or by the Institutional Animal Care and Use Committee of the University of Texas. Sample sizes were based on extensive previous experience with muscle fibrosis models. Animals were randomly allocated to experimental groups. Investigators collecting data were blinded to treatment groups.

**Fibrosis models.** CTX muscle injury was induced as described previously[28]. Mice were injected with 50 µl of 20 µM cardiotoxin (Sigma) or PBS (control) into the tibialis anterior muscle. Mice were culled and muscles harvested at 4, 8, 21 and 60 days following CTX injection. To assess muscle damage and the regenerative response following CTX injection, mice were injected intraperitoneally with Evans blue dye (25 mg kg$^{-1}$) 24 h prior to sacrifice at 8 days post CTX injection as previously described[7].

The skeletal muscle laceration model was performed as previously described[33, 34]. The mice were anesthetized using pentobarbital sodium (80 mg kg$^{-1}$) (Anpro Pharmaceuticals, Arcadia, CA, USA) delivered by intraperitoneal injection. The muscles were lacerated with a surgical blade (no. 11) at the largest diameter, through the lateral 50% of their width and 100% of their thickness. Polydioxanone suture material (PDSII 5-0, Ethicon, Somerville, NJ, USA), 4 mm in length, was placed at the medial edge of the lacerated side of each leg as a marker for the lacerated site. After the laceration was made, the skin was closed with 4-0 silk. All animals were sacrificed for evaluation of fibrosis at 21 days.

Myocardial fibrosis was induced as described previously[44]. Angiotensin II (Sigma) dissolved in sterile water was administered at 200 ng kg$^{-1}$ min$^{-1}$ for 14 days via subcutaneous Alzet osmotic minipumps (Durect Corporation). Prior to pump implantation, mice were conditioned to tail cuff blood pressure measurement for several days. Blood pressure was assessed on the day of surgery (day 0) and pumps were implanted under general anaesthesia (isoflurane in 100% oxygen). Further blood pressure measurements were taken on days 3, 7, 10 and finally on day 14 prior to sacrifice.

**Primary cell isolation and cell sorting.** Mouse skeletal muscle was excised, minced and digested in collagenase/dispase solution (DMEM/20%FBS/1%PS/0.5 mg ml$^{-1}$ of collagenase II-S and dispase (Sigma-Aldrich)) for 30 min at 37 °C with shaking (200–250 r.p.m.). An equal volume of DMEM/20%FBS/1%PS was added to halt the digestion and the total suspension was passed through a sterilized nylon mesh to remove large clumps. The suspension was then passed through a 100 µm followed by a 70 µm strainer and centrifuged (300 × *g*, 5 min). The supernatant was discarded and the pellet re-suspended in 10 ml red cell lysis buffer (Sigma) and incubated at room temperature for 1 min. An equal volume of DMEM/20%FBS/1% PS was added and the suspension centrifuged (300 × *g*, 5 min). The supernatant was discarded and the pellet re-suspended in 1 ml FACS buffer (PBS/2%FBS) and passed through a 40 µm strainer. Cells were counted using a haemocytometer using trypan blue to distinguish non-viable cells. A similar procedure was followed in the preparation of cardiac muscle. The atria and major vessels were removed and the remaining tissue was minced and digested using 200 U ml$^{-1}$ of Collagenase II (Gibco). Three serial digestions of 10 min were performed, each followed by removal and quenching of enzyme activity with DMEM/20%FBS/1%PS. Treatment of the suspension was as described for skeletal muscle.

Isolation of GFP$^+$ cells from mTmG;*Pdgfrb*-Cre reporter mice: Following live/dead staining with DAPI (Invitrogen), live single GFP$^+$ cells from mTmG;*Pdgfrb*-Cre mice were sorted using a FACSAria Cell Sorter (BD Biosciences). Fluorescence compensation settings were optimised using anti-mouse Ig, κ/negative control beads plus (BD Biosciences) incubated with GFP antibody (BD Bioscience). Unstained cells from mTmG;*Pdgfrb*-Cre negative mice were used to account for the autofluorescence of samples and an isotype (BD Bioscience) was used as a negative control. The gating strategy used to isolate GFP$^+$ cells is shown in Supplementary Fig. 11a.

Isolation of PDGFRβ$^+$ cells from non-reporter tissue: When sorting non-reporter tissues, an anti-PDGFRβ antibody was used to identify PDGFRβ$^+$ cells. In brief, cells were re-suspended at a concentration of 3.0 × 10$^7$ ml$^{-1}$ and incubated with anti-PDGFRβ antibody (phycoerythin- (PE) conjugated, clone ABP5, 1:50, no. 136006, Biolegend). As controls, 5 × 10$^5$ cells were incubated with the isotype control (PE Rat IG2bα, clone RTK27581:50, no. 400508, Biolegend) in the same conditions. Cell suspensions were washed with FACS buffer and centrifuged (300 × *g*, 5 min). The supernatant was then discarded and the cells re-suspended in 1 ml FACS buffer prior to sorting. Fluorescence compensation settings were optimised using anti-mouse Ig, κ/negative control beads plus (no. 560497, BD Biosciences) incubated with the anti-PDGFRβ antibody. Unstained cells were used to account for the autofluorescence of samples and fluorescently-matched isotypes were used as negative controls. Prior to selection of PDGFRβ$^+$ cells, a side versus forward scatter plot was used to remove debris, then a height versus width plot was used to eliminate doublets. DAPI (0.1–0.5 µg ml$^{-1}$ (Invitrogen)) was used to stain dead cells (Supplementary Fig. 11b).

**Primary cell culture.** Sorted cells were seeded onto tissue culture plates, at a density of 2 × 10$^4$ cells per cm$^2$ and cultured in high-glucose DMEM/20%FBS/1% PS in a 37 °C, 5% CO$_2$ incubator. Medium was refreshed three times per week, with cells passaged on reaching 90% confluence.

**Flow cytometry.** Cells were washed twice with PBS and re-suspended in FACS buffer. 4 × 10$^5$ cells were then incubated with a conjugated antibody (or isotype control) at 4 °C for 30 min in the dark. Cells were then washed, re-suspended in FACS buffer and analysed on a Becton Dickinson LSR Fortessa II. Antibodies used were Rat anti mouse PE-conjugated antibody to PDGFRβ (Clone APB5, 1:50, no. 136006, Biolegend) and isotype control (PE Rat IG2bα 1:50, Clone RTK2758 no. 400508, BD Bioscience); Rat anti mouse PE-conjugated antibody to αv (CD51) (Clone RMV-7, 1:200, no. 12-0512-83 eBioscience) and isotype control (PE Rat IgG1κ 1:200, no. 12-4301-83, eBioscience); Rat anti mouse AF700-conjugated antibody to CD34 (Clone RAM34, 1:100, no. 560518, BD Bioscience) and isotype control (AF700 Rat IgG2b, Clone A95-1, 1:100, no. 557964, BD Bioscience); Rat anti mouse APC-conjugated antibody to CD31 (Clone 390, 1:100, no. 102410 BioLegend) and isotype control (APC Rat IgG2a, Clone RTK2758, 1:100, no 400511, BioLegend); Rat anti mouse APC-conjugated antibody to PDGFRα (Clone APA5, 1:100, no. 562777, BD Bioscience) and isotype control (APC Rat IgG2a, Clone R35-95, 1:100, no. 553932, BD Bioscience).

**In vivo CWHM 12 and CWHM 96 studies.** For all studies CWHM 12 and CWHM 96 were solubilized in 50% DMSO (in sterile water) and dosed to 100 mg kg$^{-1}$ day$^{-1}$. Drug or vehicle (50% DMSO) were delivered by implantable Alzet osmotic minipumps (Durect). For CTX-induced fibrosis, pumps were inserted subcutaneously either before the CTX injection or ten days following the CTX injection. For laceration-induced fibrosis, pumps were inserted subcutaneously before the laceration was performed. Muscles were harvested after 21 days. For AngII-induced cardiac fibrosis, pumps were inserted subcutaneously seven days after AngII dosing commenced. Hearts were harvested 14 days after AngII dosing commenced.

**In vivo TGFβ1 rescue experiment.** Mice were treated with the small molecule inhibitor of αv integrins (CWHM12) and control from the time of CTX induced skeletal muscle injury. Mice then received recombinant TGFβ1 (50 ng, recombinant human TGFβ1; R&D Systems, Minneapolis, MN, USA) injections every 72 h to the region of CTX-induced muscle injury in a volume of 50 µl of PBS. Muscles were collected for analysis at 21 days.

**Apoptosis assay.** Cells were seeded in 6 well plates and treated with 0.1 µM CWHM 12 or CWHM 96 or 1 µM staurosporine for 6 h. Cells were lysed and assayed for cleaved caspase 3 by ELISA (Human/mouse cleaved caspase-3 (ASP-175) Cat. No. DYC835-2 R&D systems).

**Migration assay.** 1 × 10$^5$ cells in serum free media were seeded in the top of a 8 µm pore size, 5.7 mm diameter, PCTE ChemoTx disposable membrane (Neuroprobe) and media containing 10% FCS with or without 0.1 µM CWHM 12 or CWHM 96 was added to the lower chamber. Cells were allowed to migrate for 3 h and 18 h at 37 °C. Cells were wiped from the upper surface of the membrane and cells on the underside were fixed in 70% ethanol and stained with Diffquick (Fischer Scientific). Migrated cells were quantified colorimetrically at 550 nm and expressed as % of total (unwiped) cells.

**Determination of CWHM 12 and CWHM 96 in plasma and tissue.** Plasma samples (50 µl total volume) were diluted with control naive mouse plasma as appropriate to bring sample into the range of the standard curve. Tissue samples (skeletal and cardiac muscle) were homogenized at 0.2 g tissue/1 ml PBS using a mini-beadbeater (Biospec Products, Bartlesville, OK, USA). Tissue homogenates were extracted neat (0.2 g ml$^{-1}$) (50 µl total volume). Corresponding naïve control mouse matrices were used to construct individual plasma and tissue standard curves. Internal standard was added to all samples and standards at 200 ng ml$^{-1}$ (final), followed by a protein precipitation extraction with 150 µl of acetonitrile. The samples were capped and mixed on a multiplate vortexer for 5 min and centrifuged for 5 min at 3200 r.p.m. The supernatant was transferred to a 96-well sample plate and capped for LC–MS/MS analysis using a system consisting of an LC-20AD pump (Shimadzu, Kyoto, Japan), an HTC PAL autosampler (Leap technologies, Carrboro, NC, USA), and a Sciex API-4000 mass spectrometer in ESI mode (AB Sciex, Foster City, CA, USA). An Amour C18 reverse-phase column (2.1 × 30 mm, 5 µm; Analytical Sales and Services, Pompton Plains, NJ, USA) was used for chromatographic separation. Mobile phases were 0.1% formic acid (aqueous) and 100% acetonitrile (organic) with a flow rate of 0.35 ml min$^{-1}$. The starting phase was 10% acetonitrile for the 0.9 min, increased to 90% acetonitrile over 0.4 min, maintained for an additional 0.2 min, returned to 10% acetonitrile over 0.4 min, and then held for 1.6 min. The multiple reaction

monitoring (MRM) transition for CWHM 12 was as follows: mass-to-charge ($m/z$), 590.13 > 234.1. Peak areas were integrated using Analyst 1.5.1 (AB Sciex, Foster City, CA, USA).

**Immunohistochemistry and immunofluorescence.** For immunofluorescence staining, skeletal muscle tissue was fixed in 4% paraformaldehyde overnight at 4 °C, immersed in graded sucrose solutions, embedded in OCT (Tissue Tek), and flash frozen in liquid nitrogen before storage at −80 °C. The following primary antibodies were used for immunohistochemistry: PDGFRβ (Rabbit anti human/mouse clone Y92, 1:50, no. ab32570, Abcam), Myosin (Mouse anti human/mouse, Clone NOQ7.5.4D, 1:1000, no. M4276, Sigma), GR1 (Rat anti mouse, 1:750, MAB1037, R&D), F4/80 (Rat anti mouse, Clone Cl:A3-1, 1:100, no. ab6640, Abcam), Isolectin B4 (1:100, no. I21411, Life Technologies), PDGFRα (Rabbit anti mouse/human, Clone D1E1E, 1:50, no. 31745S, Cell Signalling), CD34 (Rabbit anti mouse/human, Clone EP373Y, 1:100, no. ab81289, abcam), ICAM-2 (Rat anti mouse, Clone 3C4 (mIc2/4), 1:50, no. 553326, BD Pharminogen), CD45 (Rat anti mouse, 1:50, no. MAB114R, R&D), CD144 (Rat anti mouse, Clone 11D4.1, 1:50, no. 555289, BD Pharminogen) and CD31 (Rat anti mouse, Clone MEC 13.3, 1:50, no. 550274, BD Pharminogen). Sections were washed with Dulbecco's PBS (DPBS)/Tween20 pH7.4 (2 × 5 min) then blocked for one hour with 5% goat serum. After blocking, sections were incubated with fluorochrome-conjugated antibodies overnight at 4 °C or were blocked to prevent non-specific avidin and biotin interactions prior to overnight incubation with the primary antibody at 4 °C. Sections incubated with primary antibodies were washed with DPBS/Tween20 pH 7.4 (2 × 5 min) prior to incubation with the secondary antibody for one hour at RT. All sections were then washed with DPBS/Tween20 pH 7.4 (2 × 5 min) and incubated with DAPI (5 μg ml⁻¹) and Alexa Fluor-coupled streptavidin [1:1000 (Invitrogen)] for 45 min. After a final DPBS/Tween20 pH 7.4 wash (2 × 5 min), the sections were mounted in fluorescent mounting medium (Dako) and allowed to dry for 1 hour. Confocal imaging was performed on a Leica SP8 microscope.

Percentage field coverage of GFP⁺ cells was calculated by capturing 10 random confocal images from cryosections of mouse skeletal muscle at 4, 8, 21 and 60 days after control (PBS) or CTX intramuscular injection. GFP reporting was subjected to threshold processing and calculation of percentage field coverage using ImageJ software[45]. Antibody staining for CD31, GR1 and F4/80 was quantified by capturing ten random fluorescent images from cryosections of mouse skeletal muscle. Antibody staining was subjected to threshold processing and calculation of percentage field coverage using ImageJ software. All images within an experiment were processed equally.

For assessment of cardiomyocyte cross-sectional area heart sections were stained with rhodamine labelled antibody to wheat-germ agglutinin (Vector Laboratories RL-1022) to label cell membranes and DAPI to label nuclei. Ten ×200 magnification random fields were captured per heart using a fluorescence microscope and the cross-sectional area of cells with central nuclei was calculated.

Paraffin-embedded mouse sections were stained for picrosirius red and haematoxylin and eosin. Blinded, stained sections (5 μm) were quantified using ImageJ software. For skeletal muscle, ten random fields from each section, generated using a stereologer microscope, were analysed at a final magnification of ×40. For quantification of heart fibrosis, unfixed hearts were bisected transversely at the midlevel of the left ventricle and the basal portion was fixed in 10% neutral buffered formalin. Six transverse sections, separated by approximately 50 μm, were taken from the cut surface and stained with picrosirius red and, from these, three sections, in which the papillary muscle insertions were clearly visible, were selected for assessment. Threshold analysis was carried out on each section to quantify the area occupied by collagen deposits as a percentage of the total area of the left and right ventricles and inter-ventricular septum. A mean value was generated based on the results from the three sections.

For assessment of phospho-SMAD 3 staining paraffin-embedded sections were de-parrafinized in xylene (2 × 5 min) and rehydrated through 100–50% alcohols (2 min each) before being pre-treated with 3% hydrogen peroxide for 10 min at room temperature (RT). Slides were then washed with PBS (2 × 3 min) followed by antigen retrieval with Tris-EDTA (pH 9) 20 min. Slides were then washed with PBS (2 × 3 min) and mounted in Sequenza racks before being incubated with Avidin and Biotin block (Vector, SP-2100) for 15 min each with PBS washes in between (2 × 3 min). Slides were then incubated with 20% goat serum for 30 min at RT, followed by overnight incubation with anti-phospho-SMAD3 antibody (1:100, Abcam Ab52903) at 4 °C before washing with PBS (2 × 3 min). Sections were then washed with PBS (2 × 3 min) then incubated with goat anti-rabbit biotinylated antibody (1:1000, Vector, BA-1000) for 30 min at RT followed by PBS wash (2 × 3 min). Slides were then incubated with Vectastain Elite ABC reagent (Vector, PK-7100) for 30 min and washed with PBS (2 × 3 min) before incubation in DAB substrate kit (Dako, K3468) for 10 min at RT followed by PBS wash (2 × 3 min). Slides were then counter-stained with Harris haematoxylin and Scotts tap water. For each blinded, stained section twelve random fields were generated using a stereologer microscope, and the number of DAB-positive nuclei were counted at a final magnification of ×40.

For quantification of heart fibrosis, unfixed hearts were bisected transversely at the midlevel of the left ventricle and the basal portion was fixed in 10% neutral buffered formalin. Six transverse sections, separated by approximately 50 μm, were taken from the cut surface and stained with picrosirius red and, from these, three

sections, in which the papillary muscle insertions were clearly visible, were selected for assessment. Threshold analysis was carried out on each section to quantify the area occupied by collagen deposits as a percentage of the total area of the left and right ventricles and inter-ventricular septum. A mean value was generated based on the results from the three sections.

**Assessment of recombination efficiency.** Ten confocal images (63×) were randomly acquired throughout non-adjacent cryosections of skeletal and cardiac muscle. Antibody staining for PDGFRβ and expression of GFP in skeletal muscle from mTmG:*Pdgfrb*-Cre mice were separately subjected to threshold processing using ImageJ software and then the percentage of GFP⁺ cells staining PDGFRβ⁺, and the percentage of PDGFRβ⁺ cells expressing GFP were quantified.

**TGFβ activation assay.** PDGFRβ⁺ cells isolated by FACS were cultured in DMEM/20%FBS/1%PS for 5 days on tissue culture plastic and then plated at $5 \times 10^4$ cells per well in 96-well plates with $1.5 \times 10^4$ mink lung epithelial cells (TMLCs), expressing firefly luciferase downstream of a TGFβ sensitive portion of the plasminogen activator inhibitor-1 promoter. Cells were cultured for 16 h in DMEM/20%FBS/1%PS in the presence of CWHM 12 or CWHM 96 (control), with or without the addition of recombinant human TGFβ1 (R&D) or TGFβ-blocking antibody. TGFβ activity was calculated by measurement of luminescence following cell lysis and addition of ATP/luciferin. Luminescence data was normalised by measurement of total protein levels. Recombinant human TGFβ1 was reconstituted as per the manufacturer's instructions.

**Human skeletal and cardiac muscle tissue.** The use of human tissues for this study was approved by the Local Ethics Committee at the University of Edinburgh.

**qRT-PCR.** Total RNA was isolated using an RNeasy Micro Kit (Qiagen). cDNA was analysed by SYBR-Green real-time PCR with a Roche Lightcycler and normalized to β-actin. Primers used were as follows: *Actb* forward: TGTTACC AACTGGGACGACA, *Actb* reverse: GGGGTGTTGAAGGTCTCAAA; *Itgav* forward: CCGTGGGACTTCTTCGAGCC, *Itgav* reverse: CTGTTGAATCAAACT-CAATGGGC; *Pdgfrb* forward: TCCAGGAGTGATACCAGCTTT, *Pdgfrb* reverse: CAGGAGCCATAACACGGACA; *Acta2* forward: GTCCCAGACATCAGGGA GTAA, *Acta2* reverse: TCGGATACTTCAGCGTCAGGA; *Col1a1* forward: GCTCCTCTTAGGGGCCACT, *Col1a1* reverse: CCACGTCTCACCATTGGGG; *Col 3a1* forward: AACCTGGTTTCTTCTCACCCTTC, *Col 3a1* reverse: ACTC ATAGGACTGACCAAGGTGG; *Tgfb1* forward: CTCCCGTGGCTTCTAGTGC, *Tgfb1* reverse: GCCTTAGTTTGGACAGGATCTG; *Mmp2* forward: CAAGTTCC CCGGCGATGTC, *Mmp2* reverse: TTCTGGTCAAGGTCACCTGTC; *Mmp13* forward: CTTCTTCTTGTTGAGCTGGACTC, *Mmp13* reverse: CTGTGGAGGT CACTGTAGACT; *Timp1* forward: TGCAACTCGGACCTGGTCATA, *Timp1* reverse: CGCTGGTATAAGGTGGTCTCG. *Itgb1* forward: CTACTTCTGCACG ATGTGATGAT, *Itgb1* reverse: TTGGCTGGCAACCCTTCTTT; *Itgb3* forward: CCACACGAGGCGTGAACTC, *Itgb3* reverse: CTTCAGGTTACATCGGGGTGA; *Itgb5* forward: GAAGTGCCACCTCGTGTGAA, *Itgb5* reverse: GGACCGTGGAT TGCCAAAGT; *Itgb8* forward: CTGAAGAAATACCCCGTGGA, *Itgb8* reverse: ATGGGGAGGCATACAGTCT.

**Skeletal muscle functional testing.** Mechanical assessment of the muscle tissue was performed in situ using custom mechanical testing apparatus, Aurora Scientific 1300A (Aurora Scientific, Ontario, Canada) fitted with a 1 N load cell. The control and data acquisition was achieved in LabVIEW v8.5 (National Instruments, TX, USA) using Aurora Scientific control and analysis programs DMCv5.500 and DMAv5.300 respectively. Anaesthesia was achieved using 4% isoflurane in 100% oxygen in an induction chamber and was maintained using 2% isoflurane in oxygen via nosecone during dissection and mechanical assessment. The tibialis anterior muscle was dissected and a 5-0 silk suture (Ethicon) was secured around the tendon distally. The mouse was securely held in place on a testing mat and kept warm during the mechanical evaluation by a heating lamp. Saline solution was applied via pipette to the exposed muscle to prevent dehydration during the experimental process. The leg was secured in place using a custom U-clamp attached at the proximal tibial plateau and a BD PrecisionGlide 25 G needle (BD, NJ, USA) behind the patella tendon. The silk suture was attached to the lever arm of the mechanical testing apparatus by a BD PrecisionGlide 27 G needle (BD, NJ, USA). A pre-load of 1 g was applied to the muscle by adjusting the position of the testing lever via the micromanipulator. Muscle stiffness was assessed by applying a displacement of 1 mm at a rate of 2 mm s⁻¹. The force and displacement data was recorded to PC via an Aurora Scientific 604 A signal interface (Aurora Scientific, Ontario, Canada). Mechanical evaluation was performed blinded to the treatment received.

**Statistical analyses.** All data are presented as mean ± SEM. Statistical significance was calculated using unpaired Student's $t$-tests. Differences with a $P$-value < 0.05 were considered statistically significant.

**Data availability**. The data sets generated during and/or analysed during the current study are available from the corresponding authors on reasonable request.

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

## Acknowledgements

This work was supported by a Wellcome Trust funded Edinburgh Clinical Academic Track (ECAT) Lectureship (ref. 097483) and Royal College of Surgeons of Edinburgh small research grant to I.R.M., a British Heart Foundation Centre for Vascular Regeneration Grant (ref. RM/13/2/30158) to B.P., a British Heart Foundation PhD Fellowship (ref. SRG/15/80) to J.B., and a Wellcome Trust Senior Research Fellowship in Clinical Science (ref. 103749) to N.C.H.

## Author contributions

I.R.M., J.B., B.P. and N.C.H. conceived and designed the project. I.R.M., Z.N.G., J.B., R.D., A.C.M., J.D., M.S. and M.A.C. performed the experiments with assistance from R.J.W., A.I.T., S.N.G., J.R.S. and K.P.C. I.R.M., Z.N.G., J.B., M.S., M.A.C. and N.C.H. analysed the data. D.W.G. and P.G.R. contributed the small molecule αv integrin inhibitor CWHM 12 and the control enantiomer (CWHM 96). J.P.I., G.A.G., T.J.K., Y.L., J.H. and H.S. provided intellectual contribution. I.R.M. and N.C.H. wrote the manuscript.

## Additional information

**Competing interests:** P.G.R. and D.W.G. hold equity in Antegrin Therapeutics, LLC. The remaining authors declare no competing financial interests.

