## [Peer Review file · Nature Communications]

Reviewers' comments:

Reviewer #1 (Remarks to the Author):

Murray and coworkers investigate the role of α_v integrin-mediated TGF- β activation in fibrosis of skeletal and cardiac muscle. This study follows previous works of the authors where they generated a mouse model of conditional α_v integrin knockout under control of the PDGFR β promoter. These mice are protected from kidney, liver, and lung fibrosis and exhibit low levels of active TGF- β . PDGFR β is expressed in pericyte-lineage cells and upregulated during activation into myofibroblasts in fibrosis of various organs. Knocking down integrin α_v in these cells attenuates development of fibrosis in these organs. The authors show that PDGFR β -expressing cells are also main contributors to the myofibroblast pool in heart and skeletal muscle. The study is performed at very high technical level with a wealth of high-quality data. There are very few concerns with experimental design and interpretation. However, conceptually there is little advance over previous studies. The mechanism of TGF- β activation by α_v integrins has been demonstrated for lung, kidney, and liver using the same experimental approach. Involvement of α_v integrins β_5 and β_3 in heart fibrosis has been published before. The authors do not address the remaining open question whether these integrins are indeed the main drivers of heart fibrosis or whether integrin $\alpha_v\beta_1$ will be the predominant TGF- β activating integrins, as shown in other organs. In its present state, the study may not be sufficiently original to represent a major advance in the field.

Major:

1. The question of fibroblast and myofibroblast precursors in conditions of fibrosis has been addressed by others, in particular for the heart. Lineage tracing studies and respective animals are available that define various pro-fibrotic cell populations. Ultimately, the authors fail to show which of these cells (if any) express PDGFR β and where PDGFR β stands in the hierarchy of pro-fibrotic cell lineage markers. This is not irrelevant since the authors insinuate that heart fibrosis spreads from perivascular origins to the interstitium. This view may not be shared by all members of this research community.
2. Similarly, the identity of PDGFR β expressing cells in skeletal muscle, other than being pericytes, remains also unclear and should be investigated.
3. Although the authors state that the anti-fibrotic effects of α_v integrin knockdown under PDGFR β promoter control is due to reduced levels of active TGF- β , this is not actually shown. In the mouse model of skeletal muscle fibrosis the authors use col1a1 mRNA as indicator of reduced TGF- β activity. In vitro they do not use the transgenic α_v integrin knockout cells but cells but treat PDGFR β + with an inhibitor of α_v integrin. It is acknowledged that direct demonstration of active TGF- β in vivo is difficult but pSmad staining with and without administration of TGF- β blocking antibodies (to control for TGF-specificity) could provide useful information in vivo. The authors have used osmotic pumps before for similar approaches. Administration of active TGF- β could be used as additional control that the mouse phenotype is indeed induced by lack of active TGF- β and can be rescued.
4. The authors are somewhat inconsistent with their fibrotic activation markers. Smooth muscle actin is presented at the beginning and in some in vitro experiments but col1a1 is later mostly used in the animal models for no obvious reasons.
5. With the PDGFR+ fluorescent cells at hand, the authors should identify the α_v integrins expressed in these cells in conditions of fibrosis or control, ideally without sub-culturing. Figure 4 only demonstrates general α_v expression but the β subunits are not being studied. It is actually not clear, although reasonable to assume from the conditional knockout, that the PDGFR- β + cells are the only critical α_v integrin expressers. One could sort for α_v expressing cells and determine the fraction of PDGFR β + cells in the resulting population.

Reviewer #2 (Remarks to the Author):

This manuscript explores the molecular mechanism(s) which drives fibrosis in skeletal and cardiac muscle. Using PDGFR-beta-Cre transgenic mice and focusing on integrin α v and activation of TGF-beta, they show that depletion of PDGFR-beta resulted in reduced integrin α v and protected muscle from cardiotoxin-induced muscle damage. The authors investigated the efficacy of a small molecule peptide to inhibit integrin α v also reduced fibrosis in skeletal and cardiac tissue.

This is a well written manuscript. There are several questions that should be addressed:

Major:

1. The authors use cardiotoxin to induce damage and fibrosis in muscle. The use of snake venom in these studies may not be physiological relevant which may complicate the interpretation of data. This may be a response only to cardiotoxin only. Although this system is used to study muscle damage and repair, more physiological relevant methods of damage and fibrosis should be included e.g. exercise-induced damage or crash injury.
2. Figure 2d: Hydroxyproline of the entire muscle should be used to quantify fibrosis in skeletal or heart in Figure 3j should be used in addition to the semi-quantitative area of Sirius red staining. In cardiac muscle this is particularly important due to the asymmetric nature of the heart. How do the authors know you are measuring the same regions of the heart?
3. Western blots should be included to quantify the decrease in PDGFR-beta, integrin α v and Col1A1 rather than just QRT-PCR
4. Physiological measurements of muscle contractility and cardiac function should be included in the study to demonstrate the therapeutic treatments have physiological relevance.
5. Did the authors measure levels of CWHM 12 and 96 peptide in the muscles after delivery. How long do the active and control peptide stay in cardiac or skeletal muscle? Was the active and control compounds at the peak PK/PD levels for activity?

Minor:

1. Figure 2c: A composite image of the entire cross-section of the muscle showing picrosirius staining should be presented at least in Supplemental data.
2. Immune response to the cardiotoxin may be a factor that is not discussed in the manuscript.
3. For experimental rigor, were those collecting data blinded to the treatment groups?
4. The mouse numbers used seem low for many studies. How were the number of experimental animals determined for the study? Was Power analysis used?
5. The author's should consult with a statistician as in some cases Student's t-tests for data analysis may not be appropriate.

Reviewer #3 (Remarks to the Author):

This paper by Murray. et al demonstrates the role of perivascular mesenchymal cells in contributing to cardiac and skeletal muscle fibrosis and the effects of targeting α v integrins on perivascular mesenchymal cells in modulating fibrosis. Skeletal and cardiac fibrosis is an immense clinical problem and the authors build on a body of knowledge that has shown the pro-fibrotic role of pericytes and the effects of inhibiting α v integrins in other organ systems. Overall this is a well

executed study but I would suggest that the authors clarify certain critical experimental issues (outlined below) that would greatly help in elucidating the biology behind their observations.

1)The authors do not use an inducible Cre and so in an injury model, it is possible that fibroblasts in the injured region express PDGFR β and undergo recombination. This would thus result in GFP positive fibroblasts but not necessarily derived from the perivascular mesenchymal cells. This is an inherent problem in not using an inducible Cre and the authors should address this issue and perhaps perform immunostaining to show that activated resident fibroblasts do not express PDGFR β following injury. This is particularly germane in the heart as a fraction of cardiac fibroblasts are known to express PDGFR β .

2)The reasons behind the decreased fibrosis following deletion of the α v integrins using the *Itgav* floxed mouse are not clear. Is this secondary to fibroblast apoptosis? Is it primarily an effect on secretion of collagen type 1. Is it because the migration of PDGFR β perivascular cells to the injured area is affected? Providing these insights would make for a much better manuscript.

3)Augmented angiogenesis leads to better wound healing and decreased scarring. I think the authors should examine whether deletion of α v integrins on perivascular mesenchymal cells affects neovascularization of the injured region both in the heart and skeletal muscle.

4)Ang II induces cardiac hypertrophy in addition to fibrosis. The authors should present data on the degree of cardiac hypertrophy following deletion of the α v integrins in perivascular cells. Fibrosis is thought to be a compensatory response to cardiac hypertrophy and would be interesting to see whether the inhibition of fibrosis attenuates cardiac hypertrophy.

5)Again, as alluded to above, the authors should examine the mechanisms of reduction of fibrosis using the small molecule inhibitor of α v integrins (apoptosis, collagen secretion etc)

6)The experiments performed to determine effects of small molecules against established fibrosis (Fig 4) are intriguing. The authors should examine whether the reduction in fibrosis is secondary to the inhibition of fibrosis from Day 10-21 (skeletal muscle) /Day 7-14 in heart OR is there a reversal of existing fibrosis that occurred from Day 0-7. If it is reversal of fibrosis, then the authors need to examine this in greater detail. If it is not reversal of fibrosis, then the authors should clearly state that.

Reviewer #4 (Remarks to the Author):

The manuscript by Murray et al., describes the role of *Pdgfrb* expressing cells in cardiac and skeletal muscle fibrosis. The authors demonstrate that α v integrin is expressed in *Pdgfrb* + cells in the heart and skeletal muscle. They further show that chemical inhibition of α v integrin or genetic deletion of α v integrin with a *Pdgfrb*-Cre line ameliorates cardiac and skeletal muscle fibrosis. This work extends previous publications by this group and Dean Sheppard, which initially described a role for α v integrin in *Pdgfrb*+ cells in kidney and lung fibrosis. While the findings in the current manuscript are important to the field, it essentially duplicates previous work in a new tissue type. The rationale and methodology is sound, but lacks depth in demonstrating *Pdgfrb* cell expansion in heart and skeletal muscle injury models and does not include functional data.

Major points:

1. Figure 3a shows co-localization of the GFP-reporter co-localization with endogenous PDGFR β in top panel and lining isolectin + vessels in middle panel. It is difficult to understand what the top panel is demonstrating. If it's a single mural cell it is much too large compared to the middle panels with isolectin staining.

2. The authors demonstrate expansion of Pdgfrb cells in heart and injury models by immunofluorescence. This method is not adequate in isolation, particularly using the resolution shown here. Improving the resolution / magnification of GFP reporter images would aid in showing expansion of Pdgfrb population. Fig. 1e and 3e seems to show diffuse fluorescence that may or may not be cellular. Therefore demonstrating this pattern is associated with cells (Dapi staining and co-immunofluorescence with other markers) is important. Flow cytometry would be an important complementary approach to substantiate expansion of the Pdgfrb population after injury.

3. Fig. 3e shows GFP in large vessels, which would seem to include endothelial cells and smooth muscle, thus begging the question what these cells might become after injury, or whether Pdgfrb is induced in a wider array of cells after injury. It is not possible using the constitutive Pdgfrb-Cre line to demonstrate "expansion" of this population. With the constitutive Cre line it could be that the transgene is initiated in additional cell types after injury.

4. Fig. 3g shows increased Col1a1 expression from "stimulated" Pdgfrb GFP traced cells. This experiment should be done on the GFP reporter cells isolated from AngII versus vehicle treated animals.

5. There is no description of cardiac or skeletal muscle function in injury models after integrin av inhibition (or deletion). Is there functional improvement as assessed by echocardiography or grip strength?

6. A more detailed analysis of the fate of Pdgfrb + cells after injury would improve this manuscript.

Minor points:

7. Deletion of av integrin by Pdgfrb should be demonstrated by Western blot.

REVIEWER #1:

General Comment

Murray and coworkers investigate the role of α v integrin-mediated TGF β activation in fibrosis of skeletal and cardiac muscle. This study follows previous works of the authors where they generated a mouse model of conditional α v integrin knockout under control of the PDGFR β promoter. These mice are protected from kidney, liver, and lung fibrosis and exhibit low levels of active TGF β . PDGFR β is expressed in pericyte-lineage cells and upregulated during activation into myofibroblasts in fibrosis of various organs. Knocking down integrin α v in these cells attenuates development of fibrosis in these organs. The authors show that PDGFR β -expressing cells are also main contributors to the myofibroblast pool in heart and skeletal muscle. The study is performed at very high technical level with a wealth of high-quality data. There are very few concerns with experimental design and interpretation. However, conceptually there is little advance over previous studies. The mechanism of TGF β activation by α v integrins has been demonstrated for lung, kidney, and liver using the same experimental approach. Involvement of α v integrins b5 and b3 in heart fibrosis has been published before. The authors do not address the remaining open question whether these integrins are indeed the main drivers of heart fibrosis or whether integrin α b1 will be the predominant TGF- β activating integrins, as shown in other organs. In its present state, the study may not be sufficiently original to represent a major advance in the field.

1.1 Reviewer Comment

The question of fibroblast and myofibroblast precursors in conditions of fibrosis has been addressed by others, in particular for the heart. Lineage tracing studies and respective animals are available that define various pro-fibrotic cell populations. Ultimately, the authors fail to show which of these cells (if any) express PDGFR β and where PDGFR β stands in the hierarchy of pro-fibrotic cell lineage markers. This is not irrelevant since the authors insinuate that heart fibrosis spreads from perivascular origins to the interstitium. This view may not be shared by all members of this research community.

1.1 Authors' response

We thank the reviewer for these helpful and thoughtful comments. We agree that the manuscript would be strengthened by the addition of data to demonstrate how the PDGFR β ⁺ population relates to the pro-fibrogenic cell populations previously implicated in skeletal and cardiac muscle fibrosis.

The principal markers that have been proposed to label pro-fibrotic cells in skeletal muscle include CD34 (Joe, 2010), PDGFR α (Uezumi, 2010; Heredia, 2013) and ADAM12 (Dulauroy, 2012). Skeletal muscle derived progenitors with bipotent fibro/adipogenic potential ('fibroadipogenic progenitors' or FAPS) which do not arise from the myogenic lineage have been described by several groups (Heredia, 2013; Uezumi, 2010; Joe 2010). Using FACS, Joe *et al.*, isolated CD34⁺ murine FAPS while Uezumi isolated a phenotypically and functionally equivalent PDGFR α ⁺ population and demonstrated spontaneous differentiation into fibroblasts in *in vitro* culture. The fibrogenic potential of the PDGFR α ⁺ population has also been shown *in vivo* following transplantation of GFP labelled cells into cardiotoxin injured muscle (Uezumi, 2010). Using the same model of CTX induced skeletal muscle injury employed in the present manuscript, Dulauroy *et al.* 2012, combined fate mapping with a parabiosis experiment to demonstrate that a population of collagen-producing, α SMA⁺ myofibroblasts developing following acute dermal or muscle injury are generated from tissue-resident ADAM12⁺ cells.

To establish where PDGFR β ⁺ cells stand in the hierarchy of previously recognised cell populations with pro-fibrotic potential during skeletal muscle fibrosis we performed flow cytometry and immunohistochemistry co-localisation studies with anti-CD34 and anti-PDGFR α antibodies in healthy and fibrotic skeletal muscle (Supplementary Figure 2). Using flow cytometry we demonstrated 45.8% (SD 3.8) colocalisation of CD34 with PDGFR β in healthy muscle and 79.2% (SD 1.2) co-localisation in fibrotic muscle. We demonstrated 31.1% (SD 3.6) colocalisation of PDGFR α with PDGFR β in healthy muscle and 44.8% (SD 1.2) colocalisation in injured muscle (Supplementary Figure 2). These data demonstrate that the pro-fibrotic PDGFR β ⁺ cell population significantly overlaps with previously described pro-fibrotic populations in fibrotic skeletal muscle. Unfortunately, there are currently no effective antibodies for FACS or IHC for ADAM12 and so we are unable to present co-localisation data for this marker. We also verified this with Lucie Peduto (Pasteur Institute) who is a world expert in the analysis of ADAM12 in muscle.

Populations of cells expressing the markers PDGFR α and CD34 have also been implicated in the development of cardiac fibrosis (Diaz-flores *et al.*, 2014; Moore-Morris *et al.*, 2014). During embryonic development fibroblasts are thought to originate from mesenchymal cells in the pro-epicardium (Norris *et al.*, 2008). In the fetal and neonatal heart fibroblasts are considered to develop from expansion of endogenous populations, epithelial to mesenchymal transition of the epicardium (Zhou *et al.*, 2010) and fibroblastic differentiation of bone marrow derived cells (Visconti *et al.*, 2006). In addition to the established sources of pro-fibrotic cells mentioned above perivascular stem cells are now also considered potential sources of fibroblasts following cardiac injury.

To establish where PDGFR β ⁺ cells stand in the hierarchy of previously recognised cell populations with pro-fibrotic potential in the heart we performed flow cytometry and immunohistochemistry co-localisation studies with anti-CD34 and anti-PDGFR α antibodies in healthy and fibrotic hearts (Supplementary Figure 5). Using flow cytometry we demonstrated 49.1% (SD 2.3) colocalisation of CD34 with PDGFR β in healthy cardiac muscle and 51.5% (SD 8.8) co-localisation in fibrotic cardiac muscle. We demonstrated 44.8% (SD 1.2) colocalisation of PDGFR α with PDGFR β in healthy cardiac muscle and 71.6% (SD 4.7) co-localisation in fibrotic hearts (Supplementary Figure 5). These data demonstrate that the pro-fibrotic PDGFR β ⁺ cell population significantly overlaps with the previously described pro-fibrotic populations in the heart.

References:

Diaz-Flores L, *et al.* CD34+ stromal cells/fibroblasts/fibrocytes/telocytes as a tissue reserve and a principal source of mesenchymal cells. Location, morphology, function and role in pathology. *Histol Histopathol* **29**, 831-870 (2014).

Dulauroy S, Di Carlo SE, Langa F, Eberl G, Peduto L. Lineage tracing and genetic ablation of ADAM12(+) perivascular cells identify a major source of profibrotic cells during acute tissue injury. *Nat Med* 2012;18(8):1262-70.

Heredia JE, Mukundan L, Chen FM, Mueller AA, Deo RC, Locksley RM, *et al.* Type 2 innate signals stimulate fibro/adipogenic progenitors to facilitate muscle regeneration. *Cell* 2013;153(2):376-88.

Joe AW, Yi L, Natarajan A, Le Grand F, So L, Wang J, *et al.* Muscle injury activates resident fibro/adipogenic progenitors that facilitate myogenesis. *Nat Cell Biol* 2010;12(2):153-63.

Moore-Morris T, *et al.* Resident fibroblast lineages mediate pressure overload-induced cardiac fibrosis. *J Clin Invest* 124, 2921-2934 (2014).

Uezumi A, Fukada S, Yamamoto N, Takeda S, Tsuchida K. Mesenchymal progenitors distinct from satellite cells contribute to ectopic fat cell formation in skeletal muscle. *Nat Cell Biol* 2010;12(2):14352.

Location of modification:

Results section L153-164; L298-309
Supplementary Figures 2 and 5

1.2 Reviewer Comment

Similarly, the identity of PDGFR β ⁺ expressing cells in skeletal muscle, other than being pericytes, remains also unclear and should be investigated.

1.2 Authors' response

Please see response to Comment 1.1

1.3 Reviewer Comment

Although the authors state that the anti-fibrotic effects of α v integrin knockdown under PDGFR β promoter control is due to reduced levels of active TGF β , this is not actually shown. In the mouse model of skeletal muscle fibrosis the authors use col1a1 mRNA as indicator of reduced TGF β activity. *In vitro* they do not use the transgenic α v integrin knockout cells but treat PDGFR β ⁺ cells with an inhibitor of α v integrin. It is acknowledged that direct demonstration of active TGF β *in vivo* is difficult but p-smad staining with and without administration of TGF β blocking antibodies (to control for TGF β specificity) could provide useful information *in vivo*. The authors have used osmotic pumps before for similar approaches. Administration of active TGF β could be used as additional control that the mouse phenotype is indeed induced by lack of active TGF β and can be rescued.

1.3 Authors' response

We are grateful to the reviewer for this excellent comment. As further evidence that our anti-fibrotic phenotype is secondary to reduced TGF β activation *in vivo*, we have performed an *in vivo* experiment which demonstrates that the anti-fibrotic phenotype seen in *itgav*^{flox/flox};PDGFR β -cre mice is rescued by the addition of recombinant TGF β 1. Mice were treated with the small molecule inhibitor of α v integrins (CWHM12) and control (CWHM96) from the time of CTX injury. Mice then received injection of recombinant TGF β 1 to the region of injured muscle every 72hrs up to 21 days as previously described (Pessina, 2014). Administration of active recombinant TGF β 1 rescued the anti-fibrotic effect of CWHM12, providing *in vivo* evidence that the observed anti-fibrotic effect of α v integrin blockade is mediated via a reduction in TGF β activity (Figure 2i).

Location of modification:

Results section L230-239
Figure 2i

Reference:

Pessina P, Cabrera D, Morales MG, Riquelme CA, Gutiérrez J, Serrano AL, Brandan E and Muñoz-Cánoves P. Novel and optimized strategies for inducing fibrosis in vivo: focus on Duchenne Muscular Dystrophy. *Skeletal Muscle* 2014;4:7.

1.4 Reviewer comment

The authors are somewhat inconsistent with their fibrotic activation markers. Smooth muscle actin is presented at the beginning and in some *in vitro* experiments but col1a1 is later mostly used in the animal models for no obvious reasons.

1.4 Authors' response

We thank the reviewer for these comments. Unfortunately, it is not possible to use α SMA as a fibroblast marker in skeletal muscle *in vivo* as differentiating and regenerating myofibres also express this marker (Springer, 2002; Cizkova, 2009; Babai, 1990). However, we used it as additional evidence of a transition to a myofibroblastic phenotype *in vitro* as we started with a purified (non-myogenic population). Furthermore, we also used col1a1 in the *in vitro* studies because this is the most robust marker of the end-point of the fibrotic process i.e. collagen 1 is the most highly upregulated type 1 fibrillar collagen during muscle fibrosis and therefore we used this as a robust readout of the fibrotic process.

References:

Springer ML, Ozawa CR, Blau HM. Transient production of alpha-smooth muscle actin by skeletal myoblasts during differentiation in culture and following intramuscular implantation. *Cell Motil Cytoskeleton* 2002;51(4):177-86.

Cizkova D, Soukup T, Mokry J. Nestin expression reflects formation, revascularization and reinnervation of new myofibers in regenerating rat hind limb skeletal muscles. *Cells Tissues Organs*. 2009;189(5):338-47.

Babai F, Musevi-Aghdam J, Schurch W, Royal A, Gabbiani G. Coexpression of alpha-sarcomeric actin, alpha-smooth muscle actin and desmin during myogenesis in rat and mouse embryos I. Skeletal muscle. *Differentiation*. 1990 Aug;44(2):132-42.

1.5 Reviewer comment

With the PDGFR β^+ fluorescent cells at hand, the authors should identify the α v integrins expressed in these cells in conditions of fibrosis or control, ideally without sub-culturing. Figure 4 only demonstrates general α v expression but the β subunits are not being studied. It is actually not clear, although reasonable to assume from the conditional knockout, that the PDGFR β^+ cells are the only critical α v integrin expressers. One could sort for α v expressing cells and determine the fraction of PDGFR β^+ cells in the resulting population.

1.5 Authors' response

We thank the reviewer for these excellent suggestions. We have used flow cytometry to delineate α v integrin expression profiles in multiple cell lineages in healthy skeletal and cardiac muscle (Supplementary Figure 3a-d, Supplementary Figure 6b-e), and we have also characterized α v integrin β subunit expression in freshly sorted PDGFR β^+ reporter cells from control and fibrotic skeletal and cardiac muscle (Supplementary Figure 3e, Supplementary Figure 6a).

1.5 Location of modification

Results Section L173-177; 263-269; 318-325; 343-346
Supplementary Figures 3 and 6

REVIEWER #2

General comment

This manuscript explores the molecular mechanism(s) which drives fibrosis in skeletal and cardiac muscle. Using PDGFR β -Cre transgenic mice and focusing on integrin α v and activation of TGF β , they show that depletion of PDGFR β resulted in reduced integrin α v and protected muscle from cardiotoxin-induced muscle damage. The authors investigated the efficacy of a small molecule peptide to inhibit integrin α v also reduced fibrosis in skeletal and cardiac tissue. This is a well written manuscript. There are several questions that should be addressed:

2.1 Reviewer comment

The authors use cardiotoxin to induce damage and fibrosis in muscle. The use of snake venom in these studies may not be physiological relevant which may complicate the interpretation of data. This may be a response only to cardiotoxin only. Although this system is used to study muscle damage and repair, more physiological relevant methods of damage and fibrosis should be included e.g. exercise-induced damage or crush injury.

2.1 Authors' response

We thank the reviewer for these helpful comments and suggestions, and we understand the reviewer's points regarding cardiotoxin (CTX). However, CTX is a widely used, reproducible model of skeletal muscle injury, inflammation and fibrosis (Dulauroy, 2014; Chawla, 2014; Li, 2016). Intramuscular CTX injection induces local myofibre necrosis, which is rapidly followed by recruitment of inflammatory cells, clearance of cellular debris, and regeneration of injured muscle. The lesions caused by CTX are highly reproducible and the process closely mimics the response to injury also seen following crush and freeze injury.

However, further to the reviewer's excellent suggestion of using a second model of skeletal muscle damage and repair to investigate whether α v integrin-mediated regulation of skeletal muscle fibrosis is common to different modes of skeletal muscle injury, we have now performed further experiments using a laceration model of muscle fibrosis (Ardite 2012; Bedair, 2008; Chan 2003). Pharmacologic blockade of α v integrins using CWHM12 delivered via osmotic minipumps also protected mice from laceration-induced skeletal muscle fibrosis (Figure 2k-m).

References:

Dulauroy S, Di Carlo SE, Langa F, Eberl G, Peduto L. Lineage tracing and genetic ablation of ADAM12(+) perivascular cells identify a major source of profibrotic cells during acute tissue injury. *Nat Med* 2012;18(8):1262-70.

Heredia JE, Mukundan L, Chen FM, Mueller AA, Deo RC, Locksley RM, et al. Type 2 innate signals stimulate fibro/adipogenic progenitors to facilitate muscle regeneration. *Cell* 2013;153(2):376-88.

Li H, Hicks JJ, Wang L, Oyster N, Philippon MJ, Hurwitz S, Hogan MV, Huard J. Customized platelet-rich plasma with transforming growth factor b1 neutralization antibody to reduce fibrosis in skeletal muscle. *Biomaterials* 2016;87:147e156

Bedair HS, Karthikeyan T, Quintero A, Li Y, Huard J. Angiotensin II receptor blockade administered after injury improves muscle regeneration and decreases fibrosis in normal skeletal muscle. *Am J Sports Med.* 2008 Aug; 36(8):1548-54.

Chan YS, Li Y, Foster W, Horaguchi T, Somogyi G, Fu FH, Huard J. Antifibrotic effects of suramin in injured skeletal muscle after laceration. *J Appl Physiol* (1985). 2003 Aug; 95(2):771-80.

Ardite E, Perdiguero E, Vidal B, Gutarra S, Serrano AL, Munoz-Canoves P: PAI-1-regulated miR-21 defines a novel age-associated fibrogenic pathway in muscular dystrophy. *J Cell Biol.* 2012, 196: 163-175.

Location of modification

Results section L250-255

Figure 2k-m

2.2 Reviewer comment

Figure 2d: Hydroxyproline of the entire muscle should be used to quantify fibrosis in skeletal or heart in Figure 3j should be used in addition to the semi-quantitative area of Sirius red staining. In cardiac muscle this is particular important due to the asymmetric nature of the heart. How do the authors know you are measuring the same regions of the heart?

2.2 Authors' response

We thank the reviewer for these comments. Digital morphometric analysis of picrosirius red is a widely accepted method of quantifying fibrosis across multiple human organs including skeletal muscle and heart (Ji, 2016; Deng, 2015; Pessina, 2014; Dulauroy, 2012). For both skeletal muscle and heart we took great care to ensure that the same regions of heart and skeletal muscle were analysed across samples. In the preparation of skeletal muscle this was achieved by performing the CTX injection at a specific region of the tibialis anterior muscle (at the level of the junction of the middle and proximal thirds) and then bisecting the area of maximal injury in the axial plane to ensure that sections are cut from the same region across samples. A stereologer microscope was then used to ensure field capture for quantification was entirely random. For heart muscle, a specific protocol was employed to ensure that the same regions of the heart were analysed across samples. Unfixed hearts were bisected transversely at the midlevel of the left ventricle and the basal portion was fixed in 10% neutral buffered formalin. Six transverse sections, separated by approximately 50 μm , were taken from the cut surface and stained with picrosirius red and from these three sections in which the papillary muscle insertions were clearly visible were selected for quantification. Threshold analysis was carried out on each section to quantify the area occupied by collagen deposits as percentage of the total area of the left and right ventricles and inter-ventricular septum. A mean value was generated based on the results from the three sections. We have now clarified this within the methods section.

References

Pessina P, Cabrera D, Morales MG, Riquelme CA, Gutiérrez J, Serrano AL, Brandan E and Muñoz-Cánoves P. Novel and optimized strategies for inducing fibrosis in vivo: focus on Duchenne Muscular Dystrophy. *Skeletal Muscle* 2014;4:7.

Dulauroy S, Di Carlo SE, Langa F, Eberl G, Peduto L. Lineage tracing and genetic ablation of ADAM12(+) perivascular cells identify a major source of profibrotic cells during acute tissue injury. *Nat Med* 2012;18(8):1262-70.

Ji YX, Zhang P, Zhang XJ, Zhao YC, Deng KQ, Jiang X, Wang PX, Huang Z, Li H. The ubiquitin E3 ligase TRAF6 exacerbates pathological cardiac hypertrophy via TAK1-dependent signalling. *Nat Commun* 2016;7:11267.

Deng KQ, Wang A, Ji YZ, Zhang XJ, Fang J, Zhang Y, Zhang P, Jiang X, Gao L, Zhu XY, Zhao Y, Gao L, Yang Q, Zhu XH, Wei X, Pu J, Li H. Suppressor of IKKe is an essential negative regulator of pathological cardiac hypertrophy. *Nat Commun* 2016;7:11432

Location of modification

Methods section L681-689

2.3 Reviewer comment

Western blots should be included to quantify the decrease in PDGFR β , integrin αv and Col1A1 rather than just QRT-PCR

2.3 Authors' response

We do not show a reduction in PDGFR β in our manuscript and therefore have not performed

western blotting for PDGFR β . In addition, antibody reagents for collagen are notoriously poor and this is why we have used Col1A1 RT-qPCR. Furthermore, for the assessment of α v integrin expression we have deliberately used flow cytometry based quantitation. This is the gold standard for measurement of integrin expression (Figure 2a), as this quantitatively measures the expression of the α v integrin subunit at the cell surface, and therefore we used this technique to quantitatively measure α v integrin expression rather than western blotting.

2.4 Reviewer comment

Physiological measurements of muscle contractility and cardiac function should be included in the study to demonstrate the therapeutic treatments have physiological relevance.

Authors' response

We thank the reviewer for this excellent suggestion and agree that it is important to show that α v integrin inhibition has functional benefits in the context of muscle fibrosis. To this end we have investigated the functional effects of small molecule α v integrin inhibition in the CTX-induced model of skeletal muscle fibrosis. In the setting of skeletal muscle fibrosis, stiffness 'contractures' are a major source of morbidity and loss of muscle function. We therefore used commercially available, customized muscle physiology testing equipment (1300A, Aurora Scientific, Canada) to assess the plasticity (stiffness) of the tibialis anterior muscle following CTX-induced muscle fibrosis to provide functional data with direct clinical relevance (Chapman, 2015; Willey, 2016). Tibialis anterior muscles from mice receiving CWHM96 (control) were stiffer than tibialis anterior muscles from those mice treated with the α v integrin small molecule inhibitor CWHM12. These data indicate that CWHM12 can improve skeletal muscle function following CTX induced muscle injury and demonstrate that CWHM12 is a promising therapeutic candidate to treat patients with skeletal muscle fibrosis (Figure 4h-j).

References

Chapman, M.A., R. Pichika, and R.L. Lieber, Collagen crosslinking does not dictate stiffness in a transgenic mouse model of skeletal muscle fibrosis. *J Biomech*, 2015. 48(2): p. 375-8.

Willey, J.S., et al., Angiotensin-(1-7) Attenuates Skeletal Muscle Fibrosis and Stiffening in a Mouse Model of Extremity Sarcoma Radiation Therapy. *J Bone Joint Surg Am*, 2016. 98(1): p. 48-55.

Location of modification

Results section L241-248

Figure 2j

2.5 Reviewer comment

Did the authors measure levels of CWHM 12 and 96 peptide in the muscles after delivery? How long do the active and control peptide stay in cardiac or skeletal muscle? Was the active and control compounds at the peak PK/PD levels for activity?

2.5 Authors' response

We thank the reviewer for suggesting these interesting experiments. In order to determine the tissue concentration of CWHM 12 and CWHM 96 we used osmotic minipumps to dose C57BL6 mice at 100mg/kg/day. Mice were sacrificed at day 3 when the drug had reached steady state and plasma samples were taken for compound analysis. We then perfused and harvested the hearts and removed the tibialis anterior muscles before performing

compound analysis using liquid chromatography – mass spectrometry (LC-MS/MS).

The mean CWHM-12 plasma concentration was almost identical to what we have reported in previous minipump studies (Ulmasov, 2016) (Supplementary Figure 9). The CWHM-96 levels are significantly higher than CWHM-12 (Supplementary Figure 9) which is also consistent with what we have seen in other studies. Drug enantiomers frequently exhibit differing pharmacokinetic parameters because a number of mechanisms can be stereoselective, among them target interaction, metabolic clearance, renal clearance, and protein binding.

For each compound, the tissue concentrations measured in skeletal and cardiac muscle were similar to each other, although necessarily, the units are different from the plasma measurements (Supplementary Figure 9). While the numbers tell us the total drug concentration at steady state in each tissue, they do not actually tell us what the drug concentration is at the site of action, i.e. the extracellular milieu in the immediate vicinity of the cell surface-expressed α v integrins. The concentration in this compartment may (or may not) be in equilibrium with the plasma compartment.

Reference

Ulmasov B, Neuschwander-Tetri BA, Lai J, Montyrskiv V, Bhat T, Yates MP, Oliva J, Prinsen MJ, Ruminski MJ, Griggs DW. Inhibitors of Arg-Gly-Asp-Binding Integrins Reduce Development of Pancreatic Fibrosis in Mice. *Cellular and Molecular Gastroenterology and Hepatology* 2016;2(4): 499-518

Location of modification

Results section L369-384
Supplementary Figure 9

2.6 Reviewer comment

Figure 2c: A composite image of the entire cross-section of the muscle showing picrosirius staining should be presented at least in Supplemental data.

2.6 Authors' response

We thank the reviewer for this helpful suggestion. We have now included a composite image of the entire cross-section of the picrosirius red stained muscles in Figure 2c.

Location of Modification

Figure 2c

2.7 Reviewer comment

Immune response to the cardiotoxin may be a factor that is not discussed in the manuscript.

2.7 Authors' response

We thank the reviewer for this helpful comment, and we agree that this should be discussed within the manuscript. We are not aware of any reports of specific immunogenicity relating to cardiotoxin. However, to ensure that the reduced fibrosis observed in mice following α v depletion is not related to changes in the inflammatory response, we have quantified neutrophil and macrophage infiltration following CTX injury in *itgav^{flox/flox};PDGFR β -cre* versus control mice (Supplementary Figure 4c). There was no difference in neutrophil and macrophage infiltration between the two genotypes.

Location of Modification

Results section 184-192, Supplementary Figure 4c

2.8 Reviewer comment

For experimental rigor, were those collecting data blinded to the treatment groups?

2.8 Authors' response

Yes, those collecting data were blinded to treatment groups. This has now been clarified within the methods section.

Location of modification

Methods Section L485

2.9 Reviewer comment

The mouse numbers used seem low for many studies. How were the number of experimental animals determined for the study? Was Power analysis used?

2.9 Authors' response

The mouse numbers used within this manuscript were based on power calculations from previous studies carried out by the authors using these models of muscle fibrosis. We tried to minimise the animal numbers wherever possible, but felt that this is the minimum number of mice required to demonstrate statistically significant results in our muscle fibrosis experiments.

2.10 Reviewer comment

The author's should consult with a statistician as in some cases Student's t-tests for data analysis may not be appropriate.

2.10 Authors' response

Prior to commencing this project we discussed mouse numbers, sample sizes and statistical tests with a statistician who felt that our sample sizes and statistical tests were appropriate.

REVIEWER #3

General comment

This paper by Murray. et al demonstrates the role of perivascular mesenchymal cells in contributing to cardiac and skeletal muscle fibrosis and the effects of targeting αv integrins on perivascular mesenchymal cells in modulating fibrosis. Skeletal and cardiac fibrosis is an immense clinical problem and the authors build on a body of knowledge that has shown the pro-fibrotic role of pericytes and the effects of inhibiting αv integrins in other organ systems. Overall this is a well executed study but I would suggest that the authors clarify certain critical experimental issues (outlined below) that would greatly help in elucidating the biology behind their observations.

3.1 Reviewer comment

The authors do not use an inducible Cre and so in an injury model, it is possible that fibroblasts in the injured region express PDGFR β and undergo recombination. This would thus result in GFP positive fibroblasts but not necessarily derived from the perivascular mesenchymal cells. This is an inherent problem in not using an inducible Cre and the authors should address this issue and perhaps perform immunostaining to show that activated resident fibroblasts do not express PDGFR β following injury. This is particularly germane in the heart as a fraction of cardiac fibroblasts are known to express PDGFR β .

3.1 Authors' response

We thank the reviewer for raising this excellent point and we agree that the PDGFR β ⁺ cells contributing to muscle fibrosis may be derived from non-perivascular locations, and so we have removed the term 'perivascular' from the title of our manuscript and the text to prevent misinterpretation. Furthermore, we fully agree that lineage tracing of PDGFR β ⁺ cells in the setting of skeletal and cardiac muscle fibrosis would be a very interesting set of experiments to perform, however currently there are no inducible PDGFR β -Cre mouse lines available that recombine well in skeletal and cardiac muscle. Although no inducible PDGFR β -Cre mouse line is currently available to facilitate these types of analysis, to further explore this area in depth we have investigated how the PDGFR β ⁺ population relates to the pro-fibroblastic populations of cells previously implicated in skeletal and cardiac muscle fibrosis.

The principal markers that have been proposed to label pro-fibrotic cells in skeletal muscle include CD34 (Joe, 2010), PDGFR α (Uezumi, 2010; Heredia, 2013) and ADAM12 (Dulauroy, 2012). Skeletal muscle derived progenitors with bipotent fibro/adipogenic potential ('fibroadipogenic progenitors' or FAPS) which do not arise from the myogenic lineage have been described by several groups (Heredia, 2013; Uezumi, 2010; Joe 2010). Using FACS, Joe *et al.*, isolated CD34⁺ murine FAPS while Uezumi isolated a phenotypically and functionally equivalent PDGFR α ⁺ population and demonstrated spontaneous differentiation into fibroblasts in *in vitro* culture. The fibrogenic potential of the PDGFR α ⁺ population has also been shown *in vivo* following transplantation of GFP labelled cells into cardiotoxin injured muscle (Uezumi, 2010). Using the same model of CTX induced skeletal muscle injury employed in the present manuscript, Dulauroy *et al.* 2012, combined fate mapping with a parabiosis experiment to demonstrate that a population of collagen-producing, α SMA⁺ myofibroblasts developing following acute dermal or muscle injury are generated from tissue-resident ADAM12⁺ cells.

To establish where PDGFR β ⁺ cells stand in the hierarchy of previously recognised cell populations with pro-fibrotic potential during skeletal muscle fibrosis we performed flow cytometry and immunohistochemistry co-localisation studies with anti-CD34 and anti-PDGFR α antibodies in healthy and fibrotic skeletal muscle (Supplementary Figure 2). Using flow cytometry we demonstrated 45.8% (SD 3.8) colocalisation of CD34 with PDGFR β in healthy muscle and 79.2% (SD 1.2) co-localisation in fibrotic muscle. We demonstrated 31.1% (SD 3.6) colocalisation of PDGFR α with PDGFR β in healthy muscle and 44.8% (SD 1.2) colocalisation in injured muscle (Supplementary Figure 2). These data demonstrate that the pro-fibrotic PDGFR β ⁺ cell population significantly overlaps with previously described pro-fibrotic populations in fibrotic skeletal muscle. Unfortunately, there are currently no effective antibodies for FACS or IHC for ADAM12 and so we are unable to present co-localisation data for this marker. We also verified this with Lucie Peduto (Pasteur Institute) who is a world expert in the analysis of ADAM12 in muscle.

Populations of cells expressing the markers PDGFR α and CD34 have also been implicated in the development of cardiac fibrosis (Diaz-flores *et al.*, 2014; Moore-Morris *et al.*, 2014). During embryonic development fibroblasts are thought to originate from mesenchymal cells in the pro-epicardium (Norris *et al.*, 2008). In the fetal and neonatal heart fibroblasts are considered to develop from expansion of endogenous populations, epithelial to mesenchymal transition of the epicardium (Zhou *et al.*, 2010) and fibroblastic differentiation of bone marrow derived cells (Visconti *et al.*, 2006). In addition to the established sources of pro-fibrotic cells mentioned above perivascular stem cells are now also considered potential sources of fibroblasts following cardiac injury.

To establish where PDGFR β ⁺ cells stand in the hierarchy of previously recognised cell populations with pro-fibrotic potential in the heart we performed flow cytometry and immunohistochemistry co-localisation studies with anti-CD34 and anti-PDGFR α antibodies in healthy and fibrotic hearts (Supplementary Figure 5). Using flow cytometry we demonstrated 49.1% (SD 2.3) colocalisation of CD34 with PDGFR β in healthy cardiac muscle and 51.5% (SD 8.8) co-localisation in fibrotic cardiac muscle. We demonstrated 44.8% (SD1.2) colocalisation of PDGFR α with PDGFR β in healthy cardiac muscle and 71.6% (SD 4.7) co-localisation in fibrotic hearts (Supplementary Figure 5). These data demonstrate that the pro-fibrotic PDGFR β ⁺ cell population significantly overlaps with the previously described pro-fibrotic populations in the heart.

References:

Diaz-Flores L, *et al.* CD34+ stromal cells/fibroblasts/fibrocytes/telocytes as a tissue reserve and a principal source of mesenchymal cells. Location, morphology, function and role in pathology. *Histol Histopathol* **29**, 831-870 (2014).

Dulauroy S, Di Carlo SE, Langa F, Eberl G, Peduto L. Lineage tracing and genetic ablation of ADAM12(+) perivascular cells identify a major source of profibrotic cells during acute tissue injury. *Nat Med* **2012**;18(8):1262-70.

Heredia JE, Mukundan L, Chen FM, Mueller AA, Deo RC, Locksley RM, *et al.* Type 2 innate signals stimulate fibro/adipogenic progenitors to facilitate muscle regeneration. *Cell* **2013**;153(2):376-88.

Joe AW, Yi L, Natarajan A, Le Grand F, So L, Wang J, *et al.* Muscle injury activates resident fibro/adipogenic progenitors that facilitate myogenesis. *Nat Cell Biol* **2010**;12(2):153-63.

Moore-Morris T, *et al.* Resident fibroblast lineages mediate pressure overload-induced cardiac fibrosis. *J Clin Invest* **124**, 2921-2934 (2014).

Uezumi A, Fukada S, Yamamoto N, Takeda S, Tsuchida K. Mesenchymal progenitors distinct from satellite cells contribute to ectopic fat cell formation in skeletal muscle. *Nat Cell Biol* **2010**;12(2):14352.

Location of modification:

Results section L153-164; L298-309

Supplementary Figures 2 and 5

3.2 Reviewer comment

The reasons behind the decreased fibrosis following deletion of the αv integrins using the *Itgav* floxed mouse are not clear. Is this secondary to fibroblast apoptosis? Is it primarily an effect on secretion of collagen type 1. Is it because the migration of PDGFR β perivascular cells to the injured area is affected? Providing these insights would make for a much better manuscript.

3.2 Authors' response

We thank the reviewer for these excellent suggestions. To further investigate the mechanism through which αv integrin inhibition reduces fibrosis we have now performed further experiments to investigate the influence of αv integrin blockade on the migration and apoptosis of PDGFR β ⁺ cells isolated from both skeletal muscle and heart. We found that pharmacologic blockade of αv integrins had no effect on migration of PDGFR β ⁺ cells isolated from skeletal muscle (Supplementary Figure 4f) or cardiac muscle (Supplementary Figure 8a) compared to control. Furthermore, blockade of αv integrins did not influence PDGFR β ⁺ cell apoptosis as measured by cleaved caspase activity (Supplementary Figure 4g and

Supplementary Figure 8b).

Location of modification

Results section L257-261; 335-341
Supplementary Figures 4 and 8

3.3 Reviewer comment

Augmented angiogenesis leads to better wound healing and decreased scarring. I think the authors should examine whether deletion of αv integrins on perivascular mesenchymal cells affects neovascularization of the injured region both in the heart and skeletal muscle.

3.3 Authors' response

We thank the reviewer for this excellent suggestion. We now include analysis of neovascularization in both the skeletal muscle and heart fibrosis models in Itgavflox/flox;PDGFR β -Cre mice compared to control. We quantified neovascularization by counting vessels in heart and skeletal muscle in control and Itgavflox/flox;PDGFR β -Cre mice. There was no difference in neovascularization between Itgavflox/flox;PDGFR β -Cre and control mice in the skeletal muscle and heart fibrosis models (Supplementary figures 4e and 8c).

Location of modification

Results section L190-192; L339-341
Supplementary Figures 4 and 8

3.4 Reviewer comment

AngII induces cardiac hypertrophy in addition to fibrosis. The authors should present data on the degree of cardiac hypertrophy following deletion of the αv integrins in perivascular cells. Fibrosis is thought to be a compensatory response to cardiac hypertrophy and would be interesting to see whether the inhibition of fibrosis attenuates cardiac hypertrophy.

3.4 Authors' response

We thank the reviewer for this very interesting point. We performed cardiomyocyte cross sectional area assessment on hearts from control and Itgavflox/flox;PDGFR β -Cre mice treated for 14 days with 200ng/kg/min Angiotensin II. Cardiac cross sections were stained with wheat germ agglutinin to highlight cell membranes and DAPI was used as nuclear stain. Cardiomyocytes with central nuclei were identified and the cross sectional area of these cells was calculated using image analysis software (Image J, NIH). There was no difference in cross-sectional area of cardiomyocytes in control versus Itgavflox/flox;PDGFR β -Cre mice following Angiotensin II treatment (Supplementary Figure 7)

References

Haudek SB, Cheng J, Du J, Wang Y, Hermosillo-Rodriguez J, Trial J, et al. Monocytic fibroblast precursors mediate fibrosis in angiotensin-II-induced cardiac hypertrophy. *J Mol Cell Cardiol.* 2010;49(3):499–507.

Kee HJ, Sohn IS, Nam K II, Park JE, Qian YR, Yin Z, et al. Inhibition of histone deacetylation blocks cardiac hypertrophy induced by angiotensin II infusion and aortic banding. *Circulation.* 2006;113(1):51–9.

Location of modification

Results section L329-333
Supplementary Figure 7

3.5 Reviewer comment

Again, as alluded to above, the authors should examine the mechanisms of reduction of fibrosis using the small molecule inhibitor of α integrins (apoptosis, collagen secretion etc)

3.5 Authors' response

Please see response to comment 3.2

3.6 Reviewer comment

The experiments performed to determine effects of small molecules against established fibrosis (Fig 4) are intriguing. The authors should examine whether the reduction in fibrosis is secondary to the inhibition of fibrosis from Day 10-21 (skeletal muscle) /Day 7-14 in heart OR is there a reversal of existing fibrosis that occurred from Day 0-7. If it is reversal of fibrosis, then the authors need to examine this in greater detail. If it is not reversal of fibrosis, then the authors should clearly state that.

3.6 Authors' response

We thank the reviewer for this interesting comment. In order to investigate whether the reduction in cardiac fibrosis represents inhibition of fibrosis from day 7-14 we analysed the levels of cardiac fibrosis present at day 7 of Ang II infusion. We found that cardiac fibrosis at day 7 was significantly less than peak fibrosis (day 14 endpoint). This demonstrates that α integrin inhibition from Day 7 to Day 14 in the AngII cardiac fibrosis model is resulting in abrogated progression of cardiac fibrosis (please see figure below).

Digital image quantification of picrosirius red staining in cardiac muscle during Angiotensin II (200ng/k-g/min) treatment. Data are expressed as mean \pm s.e.m. * p <0.05, **** p <0.0001, (Student's t-test).

To investigate whether the reduction in skeletal muscle fibrosis represents inhibition of fibrosis from Day 10 to Day 21 we analysed the levels of fibrosis present at Day 10 post CTX injection. Significant collagen deposition occurs early in this model as part of the acute inflammatory response to CTX. However, unlike the AngII cardiac model, where progression of fibrosis is inhibited by α integrin blockade (from Day 7 to the experimental endpoint Day 14), skeletal muscle injury secondary to CTX resulted in higher levels of PSR staining at day 10 than at day 21 (please see data below). As discussed above (response to reviewer comment 3.2), we did not find any difference in apoptosis or migration of PDGFR β ⁺ cells following α integrin blockade. It would be interesting to study in even greater depth how α integrin blockade causes accelerated regression of fibrosis in the CTX skeletal muscle injury model, however we feel that this is beyond the scope of this manuscript. We thank

the reviewer for suggesting that we clarify the above points in our manuscript, and to this end we have amended the text in both the results and discussion sections, ensuring that we do not use the term 'reversal of fibrosis', but using the term 'attenuation of fibrosis'.

Digital image quantification of picrosirius red staining in skeletal muscle following CTX injection. Data are expressed as mean \pm s.e.m. *** $p < 0.001$, **** $p < 0.0001$ (Student's t-test).

REVIEWER #4

General comment

The manuscript by Murray *et al.*, describes the role of PDGFR β expressing cells in cardiac and skeletal muscle fibrosis. The authors demonstrate that αv integrin is expressed in PDGFR β^+ cells in the heart and skeletal muscle. They further show that chemical inhibition of αv integrin or genetic deletion of αv integrin with a PDGFR β -Cre line ameliorates cardiac and skeletal muscle fibrosis. This work extends previous publications by this group and Dean Sheppard, which initially described a role for αv integrin in PDGFR β^+ cells in kidney and lung fibrosis. While the findings in the current manuscript are important to the field, it essentially duplicates previous work in a new tissue type. The rationale and methodology is sound, but lacks depth in demonstrating PDGFR β^+ cell expansion in heart and skeletal muscle injury models and does not include functional data.

4.1 Reviewer comment

Figure 3a shows co-localization of the GFP-reporter co-localization with endogenous PDGFR β in top panel and lining isolectin $^+$ vessels in middle panel. It is difficult to understand what the top panel is demonstrating. If it's a single mural cell it is much too large compared to the middle panels with isolectin staining.

4.1 Authors' response

We thank the reviewer for this helpful comment. We agree that the image demonstrating co-localisation of the GFP-reporter with endogenous PDGFR β is difficult to interpret and so we have replaced this with an improved image demonstrating that PDGFR β does not co-localise with the endothelial marker CD31 (Figure 3a).

Location of modification

Figure 3a

4.2 Reviewer comment

The authors demonstrate expansion of PDGFR β ⁺ cells in heart and injury models by immunofluorescence. This method is not adequate in isolation, particularly using the resolution shown here. Improving the resolution / magnification of GFP reporter images would aid in showing expansion of PDGFR β ⁺ population. Fig. 1e and 3e seems to show diffuse fluorescence that may or may not be cellular. Therefore, demonstrating this pattern is associated with cells (Dapi staining and co-immunofluorescence with other markers) is important. Flow cytometry would be an important complementary approach to substantiate expansion of the PDGFR β ⁺ population after injury.

4.2 Authors' response

We thank the reviewer for these excellent suggestions. We have increased the resolution / magnification of the GFP reporter images to better illustrate the expansion of the PDGFR β ⁺ population (Figure 1e). We have also now included reporter images of hearts from control and AngII treated mice that include Dapi staining (Figure 3e). In addition, we now include analysis of the expansion of the GFP⁺ reporter population by quantification of cell number in both the CTX model of skeletal muscle injury and the AngII model of cardiac fibrosis as determined by co-staining with DAPI and cell counting (Figure 1g and 3c).

Location of modification

Results section L129-130; L290
Figure 1e, 1g and 3c

4.3 Reviewer comment

Fig. 3e shows GFP in large vessels, which would seem to include endothelial cells and smooth muscle, thus begging the question what these cells might become after injury, or whether PDGFR β is induced in a wider array of cells after injury. It is not possible using the constitutive PDGFR β -Cre line to demonstrate "expansion" of this population. With the constitutive Cre line it could be that the transgene is initiated in additional cell types after injury.

4.3 Authors' response

We thank the reviewer for these helpful comments. We now include a further row of images demonstrating clear endothelial staining (CD31) without GFP⁺ colocalisation in larger vessels in both skeletal muscle (Supplementary Figure 1) and heart (Figure 3a). Unfortunately, there are no specific markers available as yet that allow clear differentiation of perivascular mesenchymal cells from vascular smooth muscle cells (VSMCs). To confirm that a population of cells expressing PDGFR β do expand following CTX induced muscle injury and AngII induced cardiac muscle fibrosis we used a knock-in GFP reporter mouse line (PDGFR β ;GFP knock-in) and quantified the expansion of GFP⁺ cells. There was a significant increase in the number of GFP⁺ cells (quantified by cell number using colocalisation with DAPI) in response to CTX induced skeletal muscle injury and AngII induced cardiac injury) in the PDGFR β ;GFP knock-in mice (Figures 1g and 3g).

4.3 Location of modification

Supplemental Figures 1g and 3g.

4.4 Reviewer comment

Fig. 3g shows increased Col1a1 expression from "stimulated" PDGFR β GFP traced cells. This experiment should be done on the GFP reporter cells isolated from AngII versus vehicle treated animals.

4.4 Authors' response

We thank the reviewer for this helpful comment. We now include data that demonstrates increased Col1a1 expression in GFP reporter cells isolated from AngII treated versus vehicle treated animals (Figure 3h).

Location of Modification

Results section L291-293

Figure 3h

4.5 Reviewer comment

There is no description of cardiac or skeletal muscle function in injury models after integrin α v inhibition (or deletion). Is there functional improvement as assessed by echocardiography or grip strength?

4.5 Reviewer comment

We thank the reviewer for this excellent suggestion and agree that it is important to show that α v integrin inhibition has functional benefits in the context of muscle fibrosis. To this end we have investigated the functional effects of small molecule α v integrin inhibition in the CTX-induced model of skeletal muscle fibrosis. In the setting of skeletal muscle fibrosis, stiffness 'contractures' are a major source of morbidity and loss of muscle function. We therefore used validated, commercially available customized muscle physiology testing equipment (1300A, Aurora Scientific, Canada) to assess the plasticity (stiffness) of the tibialis anterior muscle following CTX-induced muscle fibrosis to provide functional data with direct clinical relevance. Tibialis anterior muscles from mice receiving CWHM96 (control) were stiffer than tibialis anterior muscles from those mice treated with the α v integrin small molecule inhibitor CWHM12. These data indicate that CWHM12 can improve skeletal muscle function following CTX induced muscle injury and demonstrate that CWHM12 is a promising therapeutic candidate to treat patients with skeletal muscle fibrosis (Figure 2j).

Location of modification

Results section L241-248

Figure 2j

4.6 Reviewer comment

A more detailed analysis of the fate of PDGFR β ⁺ cells after injury would improve this manuscript.

4.6 Authors' response

We thank the reviewer for this excellent suggestion and we fully agree that lineage tracing of PDGFR β ⁺ cells in the setting of skeletal and cardiac muscle fibrosis would be a very interesting set of experiments to perform, however currently there are no inducible PDGFR β -Cre mouse lines available that recombine well in skeletal and cardiac muscle. Although no inducible PDGFR β -Cre mouse line is currently available to facilitate these types of analysis, to further explore this area in depth we have investigated how the PDGFR β ⁺ population relates to the pro-fibrogenic populations of cells previously implicated in skeletal and cardiac muscle fibrosis.

The principal markers that have been proposed to label pro-fibrotic cells in skeletal muscle include CD34 (Joe, 2010), PDGFR α (Uezumi, 2010; Heredia, 2013) and ADAM12 (Dulauroy, 2012). Skeletal muscle derived progenitors with bipotent fibro/adipogenic potential

(‘fibroadipogenic progenitors’ or FAPS) which do not arise from the myogenic lineage have been described by several groups (Heredia, 2013; Uezumi, 2010; Joe 2010). Using FACS, Joe *et al.*, isolated CD34⁺ murine FAPS while Uezumi isolated a phenotypically and functionally equivalent PDGFR α ⁺ population and demonstrated spontaneous differentiation into fibroblasts in *in vitro* culture. The fibrogenic potential of the PDGFR α ⁺ population has also been shown *in vivo* following transplantation of GFP labelled cells into cardiotoxin injured muscle (Uezumi, 2010). Using the same model of CTX induced skeletal muscle injury employed in the present manuscript, Dulauroy *et al.* 2012, combined fate mapping with a parabiosis experiment to demonstrate that a population of collagen-producing, α SMA⁺ myofibroblasts developing following acute dermal or muscle injury are generated from tissue-resident ADAM12⁺ cells.

To establish where PDGFR β ⁺ cells stand in the hierarchy of previously recognised cell populations with pro-fibrotic potential during skeletal muscle fibrosis we performed flow cytometry and immunohistochemistry co-localisation studies with anti-CD34 and anti-PDGFR α antibodies in healthy and fibrotic skeletal muscle (Supplementary Figure 2). Using flow cytometry we demonstrated 45.8% (SD 3.8) colocalisation of CD34 with PDGFR β in healthy muscle and 79.2% (SD 1.2) co-localisation in fibrotic muscle. We demonstrated 31.1% (SD 3.6) colocalisation of PDGFR α with PDGFR β in healthy muscle and 44.8% (SD 1.2) colocalisation in injured muscle (Supplementary Figure 2). These data demonstrate that the pro-fibrotic PDGFR β ⁺ cell population significantly overlaps with previously described pro-fibrotic populations in fibrotic skeletal muscle. Unfortunately, there are currently no effective antibodies for FACS or IHC for ADAM12 and so we are unable to present co-localisation data for this marker. We also verified this with Lucie Peduto (Pasteur Institute) who is a world expert in the analysis of ADAM12 in muscle.

Populations of cells expressing the markers PDGFR α and CD34 have also been implicated in the development of cardiac fibrosis (Diaz-flores *et al.*, 2014; Moore-Morris *et al.*, 2014). During embryonic development fibroblasts are thought to originate from mesenchymal cells in the pro-epicardium (Norris *et al.*, 2008). In the fetal and neonatal heart fibroblasts are considered to develop from expansion of endogenous populations, epithelial to mesenchymal transition of the epicardium (Zhou *et al.*, 2010) and fibroblastic differentiation of bone marrow derived cells (Visconti *et al.*, 2006). In addition to the established sources of pro-fibrotic cells mentioned above perivascular stem cells are now also considered potential sources of fibroblasts following cardiac injury.

To establish where PDGFR β ⁺ cells stand in the hierarchy of previously recognised cell populations with pro-fibrotic potential in the heart we performed flow cytometry and immunohistochemistry co-localisation studies with anti-CD34 and anti-PDGFR α antibodies in healthy and fibrotic hearts (Supplementary Figure 5). Using flow cytometry we demonstrated 49.1% (SD 2.3) colocalisation of CD34 with PDGFR β in healthy cardiac muscle and 51.5% (SD 8.8) co-localisation in fibrotic cardiac muscle. We demonstrated 44.8% (SD1.2) colocalisation of PDGFR α with PDGFR β in healthy cardiac muscle and 71.6% (SD 4.7) co-localisation in fibrotic hearts (Supplementary Figure 5). These data demonstrate that the pro-fibrotic PDGFR β ⁺ cell population significantly overlaps with the previously described pro-fibrotic populations in the heart.

References:

Diaz-Flores L, *et al.* CD34+ stromal cells/fibroblasts/fibrocytes/telocytes as a tissue reserve and a principal source of mesenchymal cells. Location, morphology, function and role in pathology. *Histol Histopathol* **29**, 831-870 (2014).

Dulauroy S, Di Carlo SE, Langa F, Eberl G, Peduto L. Lineage tracing and genetic ablation of ADAM12(+) perivascular cells identify a major source of profibrotic cells during acute tissue injury. *Nat Med* 2012;18(8):1262-70.

Heredia JE, Mukundan L, Chen FM, Mueller AA, Deo RC, Locksley RM, et al. Type 2 innate signals stimulate fibro/adipogenic progenitors to facilitate muscle regeneration. *Cell* 2013;153(2):376-88.

Joe AW, Yi L, Natarajan A, Le Grand F, So L, Wang J, et al. Muscle injury activates resident fibro/adipogenic progenitors that facilitate myogenesis. *Nat Cell Biol* 2010;12(2):153-63.

Moore-Morris T, et al. Resident fibroblast lineages mediate pressure overload-induced cardiac fibrosis. *J Clin Invest* 124, 2921-2934 (2014).

Uezumi A, Fukada S, Yamamoto N, Takeda S, Tsuchida K. Mesenchymal progenitors distinct from satellite cells contribute to ectopic fat cell formation in skeletal muscle. *Nat Cell Biol* 2010;12(2):143-52.

Location of modification:

Results section L153-164; L298-309

Supplementary Figures 2 and 5

4.7 Reviewer comment

Deletion of αv integrin by PDGFR β should be demonstrated by Western blot.

4.7 Authors' response

For the assessment of αv integrin expression we have deliberately used flow cytometry based quantitation. This is the gold standard for measurement of integrin expression (Figure 2a), as this quantitatively measures the expression of the αv integrin subunit at the cell surface, and therefore we used this technique to quantitatively measure αv integrin expression rather than western blotting.

Reviewers' comments:

Reviewer #1 (Remarks to the Author):

Murray and coworkers have submitted a substantially revised version of their manuscript with new data and figure additions that satisfyingly address all my previous concerns. The authors now better characterize the PDGFRb+ cell population with respect to other markers of pro-fibrotic cells in heart muscle. The results are state of the art and will be invaluable for the community to set these different populations into context. The authors also now provide convincing evidence that the anti-fibrotic effect of av deficiency/inhibition is mediated by reduced levels of active TGFb and can be reverted by adding active TGF to tissue. They well justified the use of different fibrotic markers (SMA versus collagen) but may want to add a note to the manuscript text.

Reviewer #2 (Remarks to the Author):

The authors have adequately addressed my comments.

Reviewer #3 (Remarks to the Author):

The authors have largely addressed my critiques; they did not find any major effects on cardiac hypertrophy, neovascularization and fibroblast apoptosis. the main effect appears to be secondary to attenuation of TGB1 driven effects on Collagen production.

I would suggest the authors include the following points in their discussion:

1)The authors have removed the term perivascular but perhaps can address in the discussion the likely identity of PDGFRB labeled cells.

2)The authors should also address and cite a recent paper published by Sylvia Evans group (Cell Stem Cell) demonstrating the absence of perivascular cell contribution to cardiac/skeletal muscle fibrosis. The paper does present data on some pitfalls pf the PDGRB Cre and perhaps important to address those pitfalls in light of this work.

Reviewer #4 (Remarks to the Author):

The revised manuscript by Murray et al. is a very well written and logical report on the role of Integrin av in skeletal muscle and cardiac fibrosis. The authors delete Itgav using PDGFR-Cre, which recombines in PDGFRB expressing cells, presumably mural and mesenchymal cells, although this expression domain may not be so straightforward. Mice devoid of Itgav via PDGFRB-Cre have reduced fibrosis after cardiotoxin injury and AngII infusion. The authors also demonstrate that Itgav inhibition using the chemical CWHM12 reduces fibrosis in similar settings. The authors suggest that Itgav derived from mesenchymal cells mediates TGFbeta-dependent fibrosis. Although the authors have addressed a number of the reviewer comments, a couple concerns remain.

Major points:

1. The authors were asked to better evaluate the identity of the PDGFRB-+ cells they are focusing on in the current study. Although the authors removed reference to PDGFRB + cells as mural cells, the identity is still important given the authors propose a mesenchymal source of Itgav in fibrosis. Indeed, a recent manuscript from the Evan's group (Guimaraes-Camboia et al., Cell Stem Cell

2017) suggest that PDGFRB-Cre has widespread expression during development, leading to recombination in multiple lineages other than mesenchymal cells, including bone marrow and skeletal muscle. Thus, *Pdgfrb*-Cre may be induced in "activated" fibroblasts, or perhaps is more broadly expressed. A number of markers could be used to more fully characterize this cell population in heart and skeletal muscle (e.g. *Tbx18*, *CD144*). Do the authors find PDGFRB-Cre expressed in bone marrow derived cells?

2. It is curious that cardiomyocyte hypertrophy in response to AngII infusion is unchanged in *Itgav* null mice or CWHM12 treated mice, particularly considering the level of fibrosis is reported to influence hypertrophic growth (e.g. Takeda et al., JCI 2010). In this light, functional data is lacking. It is important to provide echocardiographic data for the AngII experiment to complement the finding of reduced fibrosis. This should include the E/E' (cardiac relaxation as a direct function of fibrosis), ejection fraction and fractional shortening.

3. The authors also provide new data that suggest there is no change in vessel density, apoptosis, or PDGFRB+ cell migration. It's not clear how extensively this was evaluated. The last point in particular was tested using a boyden chamber assay, but only allowed 3 hrs for migration, which may be insufficient and is not satisfactory to rule out an effect on migration as reported.

4. The authors provide new data suggesting that mesenchymal *Itgav* drives fibrosis via activation of TGF β 1. 1. A Smad responsive luciferase reporter is slightly repressed by CWHM12 (although it does not repress rTGF β induced reporter levels). 2. rTGF β injection leads to massive fibrosis that is not reversed by CWHM12. However, it is difficult to interpret the results of this experiment in the context of *Itgav* activity. Smad / phospho-Smad staining of heart or skeletal muscle sections would complement this data.

Minor point:

5. Novelty is still somewhat modest, since this study essentially replicates previous publications by this group in kidney and lung fibrosis.

Reviewers' comments:

Reviewer #1

Reviewer comment 1.1

Murray and coworkers have submitted a substantially revised version of their manuscript with new data and figure additions that satisfyingly address all my previous concerns. The authors now better characterize the PDGFRb+ cell population with respect to other markers of pro-fibrotic cells in heart muscle. The results are state of the art and will be invaluable for the community to set these different populations into context. The authors also now provide convincing evidence that the anti-fibrotic effect of av deficiency/inhibition is mediated by reduced levels of active TGFb and can be reverted by adding active TGF to tissue. They well justified the use of different fibrotic markers (SMA versus collagen) but may want to add a note to the manuscript text.

Authors' response 1.1

We are grateful to the reviewer for this helpful suggestion. We now include the following text in the manuscript:

"In keeping with previous studies (Cizkova, 2009; Springer, 2002; Babai, 1990), differentiating and regenerating myofibres in skeletal muscle express α SMA, and so this marker was not used as a myofibroblast marker in skeletal muscle tissue sections." (Location of modification: RESULTS, page 6, lines 181-3)

References:

Cizkova D, Soukup T, Mokry J. Nestin expression reflects formation, revascularization and reinnervation of new myofibers in regenerating rat hind limb skeletal muscles. *Cells Tissues Organs* **189(5)**, 338-47 (2009).

Springer ML, Ozawa CR, Blau HM. Transient production of alpha-smooth muscle actin by skeletal myoblasts during differentiation in culture and following intramuscular implantation. *Cell Motil Cytoskeleton* **51(4)**, 177-86 (2002).

Babai F, Musevi-Aghdam J, Schurch W, Royal A, Gabbiani G. Coexpression of alpha-sarcomeric actin, alpha-smooth muscle actin and desmin during myogenesis in rat and mouse embryos I. *Differentiation* **44(2)**, 132-42 (1990).

Reviewer #2

The authors have adequately addressed my comments.

Reviewer #3

Reviewer comment 3.1

The authors have largely addressed my critiques; they did not find any major effects on cardiac hypertrophy, neovascularization and fibroblast apoptosis. the main effect appears to be secondary to attenuation of TGB1 driven effects on Collagen production.

I would suggest the authors include the following points in their discussion:

1)The authors have removed the term perivascular but perhaps can address in the discussion the likely identity of PDGFRB labeled cells.

Authors response 3.1

We are grateful to the reviewer for this helpful suggestion. We now include the following text within the discussion section:

"In keeping with widespread use of PDGFR β as a pericyte marker (Crisan, 2008; Dar, 2012), we found that

PDGFR β -Cre labelled perivascular cells in uninjured skeletal and cardiac muscle. However, the co-labelling of PDGFR β with PDGFR α^+ and CD34 $^+$ populations indicate that the pro-fibrotic PDGFR β^+ cell population in skeletal and cardiac muscle significantly overlaps with the previously described pro-fibrotic populations including fibro-adipogenic progenitors (FAPs) (Heredia, 2013; Uezumi, 2011) and cardiac fibroblasts (Moore-Morris, 2013; Diaz-Flores, 2014).” (Location of modification: DISCUSSION, page 13, lines 446-452)

References:

Crisan M, Yap S, Casteilla L, Chen CW, Corselli M, Park TS, Andriolo G, Sun B, Zheng B, Zhang L, Norotte C, Teng PN, Traas J, Schugar R, Deasy BM, Badylak S, Buhning HJ, Giacobino JP, Lazzari L, Huard J, Péault B. A perivascular origin for mesenchymal stem cells in multiple human tissues. *Cell Stem Cell*. 2008;3(3):301-13.

Dar A, Domev H, Ben-Yosef O, Tzukerman M, Zeevi-Levin N, Novak A et al. Multipotent vasculogenic pericytes from human pluripotent stem cells promote recovery of murine ischemic limb. *Circulation* **125(1)**, 87–99 (2012).

Heredia JE, et al. Type 2 innate signals stimulate fibro/adipogenic progenitors to facilitate muscle regeneration. *Cell* **153**, 376-388 (2013).

Uezumi A, et al. Fibrosis and adipogenesis originate from a common mesenchymal progenitor in skeletal muscle. *J Cell Sci* **124**, 3654-3664 (2011).

Moore-Morris T, et al. Resident fibroblast lineages mediate pressure overload-induced cardiac fibrosis. *J Clin Invest* **124**, 2921-2934 (2014).

Diaz-Flores L, et al. CD34 $^+$ stromal cells/fibroblasts/fibrocytes/telocytes as a tissue reserve and a principal source of mesenchymal cells. Location, morphology, function and role in pathology. *Histol Histopathol* **29**, 831-870 (2014).

Reviewer comment 3.2

2)The authors should also address and cite a recent paper published by Sylvia Evans group (Cell Stem Cell) demonstrating the absence of perivascular cell contribution to cardiac/skeletal muscle fibrosis. The paper does present data on some pitfalls of the PDGFR β Cre and perhaps important to address those pitfalls in light of this work.

Authors' response 3.2

We are grateful to the reviewer for this helpful comment. We now include the following new data in the results section and new text in the discussion section:

“Of note a recent paper by Guimarães-Camboa *et al.* (Guimarães-Camboa, 2017) suggested that PDGFR β -Cre extensively labels multiple cellular lineages including endothelial and haematopoietic lineages in cardiac and skeletal muscle. To investigate the discrepancy between our data using PDGFR β -Cre in skeletal and cardiac muscle and the data published by Guimarães-Camboa *et al.*, we stained skeletal and cardiac muscle from mTmG;PDGFR β -Cre reporter mice with the hematopoietic marker CD45 and found no co-localisation between PDGFR β reporter cells and CD45 (Supplementary Figures 1 and 5). We also stained skeletal and cardiac muscle from mTmG;PDGFR β -Cre reporter mice with the endothelial markers CD31, CD144 and ICAM-2 and demonstrated no colocalisation of PDGFR β reporter cells with CD31, CD144 and ICAM-2 (Supplementary Figures 1 and 5). Furthermore, data published by Guimarães-Camboa *et al.* (Guimarães-Camboa, 2017) suggested that recombination of myofibres was widespread in skeletal muscle in PDGFR β -Cre reporter mice. However, in marked contrast we have not seen widespread recombination of myofibres in PDGFR β -Cre reporter mice, as evidenced by 97.5% (SEM 0.46) of cells staining for PDGFR β reporting GFP, and 95.3% (SEM 1.08) of GFP $^+$ cells staining positively for PDGFR β (Figures 1b-c). Virtually all GFP $^+$ cells sorted from mTmG;PDGFR β -Cre mouse skeletal muscle stained positively with anti-PDGFR β antibody on flow cytometric analysis (Figure 1d). Furthermore, we demonstrate a lack of co-localisation of PDGFR β reporter cells with anti-myosin staining in skeletal muscle of PDGFR β -Cre reporter mice (Supplementary Figure 1), and as evidenced by

multiple images throughout our manuscript we do not see widespread recombination of myofibres in skeletal muscle of PDGFR β Cre reporter mice.” (Location of modification: DISCUSSION, page 13, lines 454-476)

Reference:

Guimarães-Camboa N, Cattaneo P, Sun Y, Moore-Morris T, Gu Y, Dalton ND, Rockenstein E, Masliah E, Peterson KL, Stallcup WB, Chen J, Evans SM. Pericytes of Multiple Organs do not behave as mesenchymal stem cells in vivo. *Cell Stem Cell* **20(3)**, 345-359 (2017).

Reviewer #4

The revised manuscript by Murray et al. is a very well written and logical report on the role of Integrin α v in skeletal muscle and cardiac fibrosis. The authors delete Itgav using PDGFR-Cre, which recombines in PDGFR β expressing cells, presumably mural and mesenchymal cells, although this expression domain may not be so straightforward. Mice devoid of Itgav via PDGFR β -Cre have reduced fibrosis after cardiotoxin injury and AngII infusion. The authors also demonstrate that Itgav inhibition using the chemical CWHM12 reduces fibrosis in similar settings. The authors suggest that Itgav derived from mesenchymal cells mediates TGF β -dependent fibrosis. Although the authors have addressed a number of the reviewer comments, a couple concerns remain.

4.1 Reviewer comment

1. The authors were asked to better evaluate the identity of the PDGFR β -+ cells they are focusing on in the current study. Although the authors removed reference to PDGFR β + cells as mural cells, the identity is still important given the authors propose a mesenchymal source of Itgav in fibrosis. Indeed, a recent manuscript from the Evan’s group (Guimaraes-Camboa et al., *Cell Stem Cell* 2017) suggest that PDGFR β -Cre has widespread expression during development, leading to recombination in multiple lineages other than mesenchymal cells, including bone marrow and skeletal muscle. Thus, Pdgfrb-Cre may be induced in “activated” fibroblasts, or perhaps is more broadly expressed. A number of markers could be used to more fully characterize this cell population in heart and skeletal muscle (e.g. Tbx18, CD144). Do the authors find PDGFR β -Cre expressed in bone marrow derived cells?

4.1 Authors’ Response

We are grateful to the reviewer for this helpful comment. We now include the following new data in the results section and new text in the discussion session:

“Of note a recent paper by Guimarães-Camboa *et al.* (Guimarães-Camboa, 2017) suggested that PDGFR β -Cre extensively labels multiple cellular lineages including endothelial and haematopoietic lineages in cardiac and skeletal muscle. To investigate the discrepancy between our data using PDGFR β -Cre in skeletal and cardiac muscle and the data published by Guimarães-Camboa *et al.*, we stained skeletal and cardiac muscle from mTmG;PDGFR β -Cre reporter mice with the hematopoietic marker CD45 and found no co-localisation between PDGFR β reporter cells and CD45 (Supplementary Figures 1 and 5). We also stained skeletal and cardiac muscle from mTmG;PDGFR β -Cre reporter mice with the endothelial markers CD31, CD144 and ICAM-2 and demonstrated no colocalisation of PDGFR β reporter cells with CD31, CD144 and ICAM-2 (Supplementary Figures 1 and 5). Furthermore, data published by Guimarães-Camboa *et al.* (Guimarães-Camboa, 2017) suggested that recombination of myofibres was widespread in skeletal muscle in PDGFR β -Cre reporter mice. However, in marked contrast we have not seen widespread recombination of myofibres in PDGFR β -Cre reporter mice, as evidenced by 97.5% (SEM 0.46) of cells staining for PDGFR β reporting GFP, and 95.3% (SEM 1.08) of GFP⁺ cells staining positively for PDGFR β (Figures 1b-c). Virtually all GFP⁺ cells sorted from mTmG;PDGFR β -Cre mouse skeletal muscle stained positively with anti-PDGFR β antibody on flow cytometric analysis (Figure 1d). Furthermore, we demonstrate a lack of co-localisation of PDGFR β reporter cells with anti-myosin staining in skeletal muscle of PDGFR β -Cre reporter mice (Supplementary Figure 1), and as evidenced by multiple images throughout our manuscript we do not see widespread recombination of myofibres in skeletal muscle of PDGFR β Cre reporter mice.” (Location of modification: DISCUSSION, page 13, lines 454-476)

Reference:

Guimarães-Camboa N, Cattaneo P, Sun Y, Moore-Morris T, Gu Y, Dalton ND, Rockenstein E, Masliah E, Peterson KL, Stallcup WB, Chen J, Evans SM. Pericytes of Multiple Organs do not behave as mesenchymal stem cells in vivo. *Cell Stem Cell* **20(3)**, 345-359 (2017).

4.2 Reviewer comment

2. It is curious that cardiomyocyte hypertrophy in response to AngII infusion is unchanged in Itgav null mice or CWHM12 treated mice, particularly considering the level of fibrosis is reported to influence hypertrophic growth (e.g. Takeda et al., JCI 2010). In this light, functional data is lacking. It is important to provide echocardiographic data for the AngII experiment to complement the finding of reduced fibrosis. This should include the E/E' (cardiac relaxation as a direct function of fibrosis), ejection fraction and fractional shortening.

4.2 Authors' Response

We thank the reviewer for this comment. We have previously performed high frequency ultrasound examination on control mice and Itgavflox/flox;PDGFR β Cre mice following 14 days treatment with vehicle or 200ng/kg/min Angiotensin II (AngII). No statistically significant differences in left ventricular ejection fraction or left ventricular fractional shortening were detected between control mice and Itgavflox/flox;PDGFR β Cre mice in either the control vehicle or AngII groups (please see figure below). We did not measure E/E' (cardiac relaxation as a function of fibrosis) as doppler assessments were not included in the cardiac ultrasound protocol at the time of this study.

We think these results may be secondary to a potential lack of sensitivity of high frequency ultrasound in the AngII model of cardiac fibrosis, as has been observed in a number of previous studies. For example, a previous study in which C57/BL6 mice were given AngII for 21 days did not demonstrate a significant difference in left ventricular diastolic or systolic internal diameter (Ichihara, 2001). In another study C57/BL6 mice were given AngII via subcutaneous osmotic pumps for 6 weeks. This resulted in a statistically significant increase in ejection fraction of approximately 10% but no differences in end diastolic left ventricular volume or diameter (Haudek, 2010). However, we have shown significant reductions in the levels of cardiac fibrosis in Itgavflox/flox;PDGFR β Cre mice compared to control, and also mice treated with CWHM12 in both prophylactic and therapeutic settings. Furthermore, we have shown in our skeletal muscle fibrosis model that skeletal muscle stiffness was significantly reduced in mice treated with CWHM12 compared to control (Figure 2j), demonstrating that αv integrin blockade using CWHM12 can improve muscle function following CTX-induced skeletal muscle fibrosis.

References:

Ichihara S, Senbonmatsu T, Price EJ, Ichiki T, Gaffney FA, Inagami T. Angiotensin II type 2 receptor is essential for left ventricular hypertrophy and cardiac fibrosis in chronic angiotensin II-induced hypertension. *Circulation* **104(3)**, 346–51 (2001).

Haudek SB, Cheng J, Du J, Wang Y, Hermosillo-Rodriguez J, Trial J, et al. Monocytic fibroblast precursors mediate fibrosis in angiotensin-II-induced cardiac hypertrophy. *J Mol Cell Cardiol* **49(3)**, 499–507 (2010).

4.3 Reviewer's comment

3. The authors also provide new data that suggest there is no change in vessel density, apoptosis, or PDGFR β + cell migration. It's not clear how extensively this was evaluated. The last point in particular was tested using a boyden chamber assay, but only allowed 3 hrs for migration, which may be insufficient and is not satisfactory

to rule out an effect on migration as reported.

4.3 Authors' Response

We have performed further migration experiments and found that pharmacologic blockade of αv integrins had no effect on migration of PDGFR β^+ cells isolated from skeletal muscle after allowing 18hrs for migration. We have included this new data in a revised version of Supplementary Figure 4.

4.4 Reviewer's comment

4. The authors provide new data suggesting that mesenchymal Itgav drives fibrosis via activation of TGFbeta1.
1. A Smad responsive luciferase reporter is slightly repressed by CWHM12 (although it does not repress rTGFb induced reporter levels). 2. rTGFb injection leads to massive fibrosis that is not reversed by CWHM12. However, it is difficult to interpret the results of this experiment in the context of Itgav activity. Smad / phospho-Smad staining of heart or skeletal muscle sections would complement this data.

4.4 Authors' Response

We have attempted to examine phospho-smad staining in this context, but have found p-SMAD immunohistochemical staining to be poor / unreliable in these tissues. However, we have shown using multiple different approaches and readouts, both *in vitro* and *in vivo*, that αv integrins on PDGFR β^+ cells regulate skeletal and cardiac muscle fibrosis, at least in part, via the activation of TGF β (Figures 2f, 2h, 2i, 4i and 4j).

4.5 Reviewer's comment

Minor point:

5. Novelty is still somewhat modest, since this study essentially replicates previous publications by this group in kidney and lung fibrosis.

4.5 Author's Response

We feel the findings in our manuscript are novel, as they identify and characterise in depth important cellular and molecular targets for the treatment of both skeletal muscle and cardiac fibrosis, which are major causes of morbidity and mortality worldwide.

Reviewers' comments:**Reviewer #4 (Remarks to the Author):**

The authors present a very thorough and well-written manuscript demonstrating the role of Itgav in muscle fibrosis. The authors have addressed all of the prior concerns, except they still fail to adequately link Itgav and the CWHM12 compound to TGFbeta signaling. Indeed, their major argument stems from recombinant TGFbeta and blocking antibody experiments, which may elicit too robust of a response to alter with Itgav modulation.

The authors report a lack of success immunostaining histological sections for total and phosphor-Smad2/3 in muscle tissue. However, this should work quite well by Western on lysates or in immunocytochemistry. Perhaps this could be done in parallel to their PAI-1 luciferase assays, especially since this reporter can be induced by multiple factors. Alternatively, the authors could tone down the claim that the effect is via TGFbeta inhibition.

Minor point:

The authors should note in the methods how the luciferase data was normalized to account for alterations in cell number/viability (e.g. total protein levels).

Reviewers' comments:

Reviewer #4

Reviewer comment 1

The authors present a very thorough and well-written manuscript demonstrating the role of Itgav in muscle fibrosis. The authors have addressed all of the prior concerns, except they still fail to adequately link Itgav and the CWHM12 compound to TGF β signalling. Indeed, their major argument stems from recombinant TGF β and blocking antibody experiments, which may elicit too robust of a response to alter with Itgav modulation. The authors report a lack of success immunostaining histological sections for total and phospho-Smad2/3 in muscle tissue. However, this should work quite well by Western on lysates or in immunocytochemistry. Perhaps this could be done in parallel to their PAI-1 luciferase assays, especially since this reporter can be induced by multiple factors. Alternatively, the authors could tone down the claim that the effect is via TGF β inhibition.

Authors' response 1

We thank the reviewer for this helpful comment. We contacted a number of groups who have expertise in immunohistochemical staining for phospho-Smad3, and we are pleased to say we have now successfully performed phospho-Smad3 immunohistochemical staining on skeletal muscle from control versus CWHM12 treated mice, which the reviewer had originally requested. Phospho-smad3 positive cells were reduced in skeletal muscle from CWHM 12 treated mice versus control following CTX injection. We have added this new data in Figure 2 (k,l). Taken together with the in vitro and in vivo data already presented (Figures 2f, 2h, 2i, 2j and 4j), these results strongly suggest that the protection from skeletal muscle fibrosis observed following α v integrin blockade is at least in part a consequence of reduced TGF- β activation.

Location of modification: Page 7 Line 236 – 244 and Figure 2k,l

Reviewer comment 2

Minor point: The authors should note in the methods how the luciferase data was normalized to account for alterations in cell number/viability (e.g. total protein levels).

Authors' response 2

We are grateful to the reviewer for this helpful comment. We have now clarified this in the methods section.

Location of modification: Page 22 Line 769

REVIEWERS' COMMENTS:

Reviewer #4 (Remarks to the Author):

The authors have addressed all of my previous concerns.